# Likely accelerated weakening of Atlantic overturning circulation emerges in optimal salinity fingerprint

Chenyu Zhu [1,2] ✉, Zhengyu Liu [3,4] ✉, Shaoqing Zhang [1,2] ✉ & Lixin Wu [1,2]

The long-term response of the Atlantic meridional overturning circulation (AMOC) to anthropogenic forcing has been difficult to detect from the short direct measurements available due to strong interdecadal variability. Here, we present observational and modeling evidence for a likely accelerated weakening of the AMOC since the 1980s under the combined forcing of anthropogenic greenhouse gases and aerosols. This likely accelerated AMOC weakening signal can be detected in the AMOC fingerprint of salinity pileup remotely in the South Atlantic, but not in the classic warming hole fingerprint locally in the North Atlantic, because the latter is contaminated by the "noise" of interdecadal variability. Our optimal salinity fingerprint retains much of the signal of the long-term AMOC trend response to anthropogenic forcing, while dynamically filtering out shorter climate variability. Given the ongoing anthropogenic forcing, our study indicates a potential further acceleration of AMOC weakening with associated climate impacts in the coming decades.

The response of Atlantic meridional overturning circulation (AMOC) to global warming has profound impact on global climate[1]. Modeling studies have long suggested an AMOC weakening response to global warming induced by anthropogenic greenhouse gasses (GHGs)[2-4]. However, due to the presence of strong interdecadal variability and the shortness of direct measurements available[5-8], whether this AMOC weakening has emerged in the real world has to be examined from a longer-term perspective, which can only be made indirectly with AMOC fingerprints. AMOC fingerprints used so far are located over the North Atlantic[9-18], most of which are not specifically designed for detecting long-term trend[9-12]. Notably, however, the classical 'warming hole' fingerprint ($T_{NA}$, Fig. 1e, Methods), which is characterized by a surface cooling over the subpolar North Atlantic (SPNA) relative to background global warming in response to reduced northward AMOC heat transport, has been used as an "all-climate" fingerprint to infer AMOC trend and variability[13,14]. In both CMIP5 and CMIP6 models, the forced responses of $T_{NA}$ do exhibit long-term trends largely consistent with multi-model ensemble mean (MMEM) AMOC (Fig. 1a, e), consistent with previous model analysis for $T_{NA}$ as an AMOC fingerprint[13,14].

An application of $T_{NA}$ to the real world has been interpreted to show an AMOC weakening trend starting from the 1950s[13,14] (Fig. 1e). However, this $T_{NA}$ weakening is reversed to strengthening from the 1990s to 2010s, largely overwhelmed by the 60-70-year Atlantic Multidecadal Oscillation (AMO, Fig. 1e, Methods) that has been identified in various fingerprints[15,19]. This dramatic difference between the observed $T_{NA}$ and model forced $T_{NA}$ response (Fig. 1e) likely suggests that the $T_{NA}$ in the real world contains strong "noise" of interdecadal variability, making it difficult to detect the global warming trend signal. Another challenge for $T_{NA}$ is the strong sensitivity of SPNA temperature to natural (volcanic) and anthropogenic aerosol radiative forcing, especially for the period of the 1950s–1990s[20-22] (Fig. 1e, f). In addition, modeling studies suggest that the relationship between $T_{NA}$ and AMOC can be nonstationary and dependent on the forcing and processes that drive AMOC changes[18,23,24].

The sparseness in AMOC measurements and the challenges on the $T_{NA}$ fingerprint lead to the following questions: (1) Has the anthropogenic AMOC weakening emerged in reality? (2) Is there an optimal fingerprint that can detect the AMOC trend more clearly than $T_{NA}$?

[1]Frontier Science Center for Deep Ocean Multispheres and Earth System (FDOMES) and Physical Oceanography Laboratory, Ocean University of China, Qingdao, China. [2]Laoshan Laboratory, Qingdao, China. [3]Atmospheric Science Program, Department of Geography, Ohio State University, Columbus, OH 43210, USA. [4]College of Geography Science, Nanjing Normal University, Nanjing, China. ✉e-mail: zhuouc@163.com; liu.7022@osu.edu; szhang@ouc.edu.cn

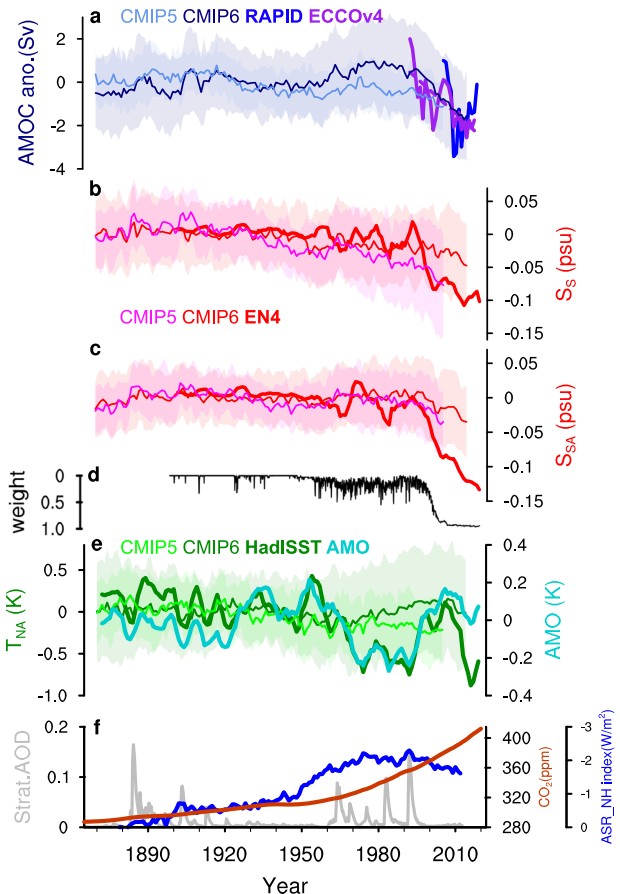

**Fig. 1 | Data-model comparison.** Time series of observed (thick lines) and multi-model ensemble mean (MMEM, thin lines; shading shows one standard deviation) (**a**) anomaly of Atlantic meridional overturning circulation (AMOC) intensity, (**b**) the salinity pileup fingerprint $S_S$, (**c**) the salinity pileup fingerprint $S_{SA}$, (**d**) monthly mean observation weight of salinity analysis over the subtropical South Atlantic (analysis value is more influenced by observations with weight closer to one), (**e**) the classical warming hole fingerprint $T_{NA}$ and (**f**) key external forcing of $CO_2$ (from NASA GISS, brown), anthropogenic (ASR_NH, blue, Methods) and volcanic (shown as global stratospheric aerosol optical depths at 550 nm; from NASA GISS, grey) aerosols. Observed fingerprints (EN4 analysis data for $S_S$ and $S_{SA}$; HadISST dataset for $T_{NA}$) are shown as 5-year running means. The AMOC anomaly from ECCOv4r4 reanalysis and RAPID measurements are shown in thick purple and blue lines with 1 and 2 offsets, respectively. The linear change of ECCOv4r4 AMOC is plotted in dashed purple line. Also shown in (**d**) is the observed AMO index (Methods). Model results are relative to the means of 1900–1950.

## Results

### Accelerated AMOC weakening after the 1980s in models

While models suggest an AMOC weakening response to increased anthropogenic GHGs[2–4], models also suggest an AMOC strengthening in response to anthropogenic aerosol cooling[4,20,25–27]. Before the 1980s, the modest GHGs forcing competes with the aerosol forcing, leaving a modest global warming[3,4] and some uncertainty in the forced AMOC response, as seen in a weak decreasing trend in CMIP5 MMEM (−0.1 Sv/50years in 1920–1980) but a weak increasing trend in the CMIP6 MMEM (0.7 Sv/50years in 1920–1980) (Fig. 1a). The difference between the two MMEMs is likely caused by aerosol forcing, which remains highly uncertain[28] and most likely overestimated in CMIP6[27,29] (Methods). In contrast, after the 1980s, the GHGs continued increasing while the aerosol forcing started decreasing[4,30] (Fig. 1f), forcing an accelerated global sea surface warming at a rate of ~0.8 °C/50 years in model simulations (~0.1 °C/50 years in 1920–1980) and likely also in observations[3,4]. Along with the unprecedented warming rate, an accelerated AMOC weakening emerged since the 1980s in both CMIP5

MMEM (−1.2 Sv/50years in 1980–2005) and CMIP6 MMEM (−4.0 Sv/50 years in 1980-2014) (Fig. 1a), which can be also detected in ~80% of individual members (Fig. S1). Interestingly, this post-1980s AMOC weakening trend appears consistent with the AMOC observations and reanalyses after the 1990s[31], including ECCOv4r4 estimate[32] (Fig. 1a), GloSea5 reanalysis[33] and a South Atlantic reconstruction[34], as well as the direct RAPID measurements at 26.5°N since 2004[5–8] (Fig. 1a, Fig. S2b).

### Optimal fingerprint for forced AMOC change

The classic AMOC fingerprint $T_{NA}$ potentially contains strong "noise" of interdecadal variability, complicating the detection of long-term AMOC trend. Indeed, this "noise" effect can also be seen in the model historical runs in the trend correlation between $T_{NA}$ and AMOC for the long period of 1850-1985 by comparing cross-member and cross-model correlations[27], with the latter favoring forced response because the ensemble mean in each model (~4 members for CMIP5 model and ~9 members for CMIP6 model, Table S1) suppresses internal variability. It is seen that the high cross-member trend correlation (~0.6) is largely reduced in the cross-model trend correlation for both CMIP5 (negative trend, to 0.02) and CMIP6 (positive trend, to 0.34) (Fig. 2e, f). Therefore, $T_{NA}$ cannot serve as a reliable fingerprint for long-term forced AMOC response.

Alternatively, we will show below that a salinity-based fingerprint firstly proposed in our previous study[35] can be a better choice for detecting this long-term forced AMOC change. This potentially optimal AMOC fingerprint is characterized by a salinity pileup over the subtropical South Atlantic (STSA), which is associated with the anomalous salinity convergence caused by AMOC changes[35] (Fig. 1b; Methods). This salinity fingerprint is defined as the sea surface salinity (SSS) difference between the STSA and the subtropical South Indo-Pacific (STSIP) $S_S = -(S_{STSA} - S_{STSIP})$, or approximately as $S_S \sim S_{SA} = -S_{STSA}$, with the minus sign to ensure the index change consistent with the sign of AMOC change. In contrast to all previous AMOC fingerprints that are located locally in the North Atlantic, our SSS fingerprint is unique in that it represents a remote AMOC response to the buoyancy flux forcing in the SPNA deep-convection region, which is known to force AMOC signal propagating southward coherently[36–38]. In response to a weakening AMOC, the northward salinity transport in the upper South Atlantic is reduced. Given the increase of mean climatological salinity from the subpolar to subtropics, the reduction in salinity transport is greater downstream (northern side of the STSA domain) than upstream (southern side of the STSA domain) in the upper branch of the AMOC, leading to a salinity pile-up in the South Atlantic[35]. The forced response of the SSS fingerprint as shown in MMEM (Fig. 1b, c) is overall consistent with that of AMOC (Fig. 1a) in both CMIP5 and CMIP6, consisting of a weak trend prior to ~1980s and an accelerated weakening afterwards. For the real world, the observed $S_S$ or $S_{SA}$ shows a slow decreasing trend in EN4 data[39] prior to 1990s (albeit uncertain in the Ishii data[40], which is too short), but a clearly accelerated weakening after the 1990s in both EN4 and Ishii data sets (Fig. 1b, c, Fig. S2a). We note that salinity observations in the earlier period in the South Atlantic tend to be sparse (with lower observation weight especially before the 1950s; Fig. 1d) and thus less reliable. Nevertheless, the lack of variability in the earlier period appears in both our salinity fingerprint and $T_{NA}$, indicating that the stronger trend/variability in the later period relative to the earlier one is possibly still valid. Moreover, in contrast to the strong oscillation in $T_{NA}$ after the 1950s, the observed SSS indices ($S_S$ and $S_{SA}$) decrease more smoothly and is less distorted by strong variability, notably, the AMO after the 1950s. As such, the observed SSS indices resemble the forced model AMOC trend signal (Fig. 1b) more than $T_{NA}$ and contains less variability noise. This visual impression can be quantified by defining the long-term linear trend as the "signal" and the remaining variability as "noise". The signal/noise ratio thus calculated for the SSS

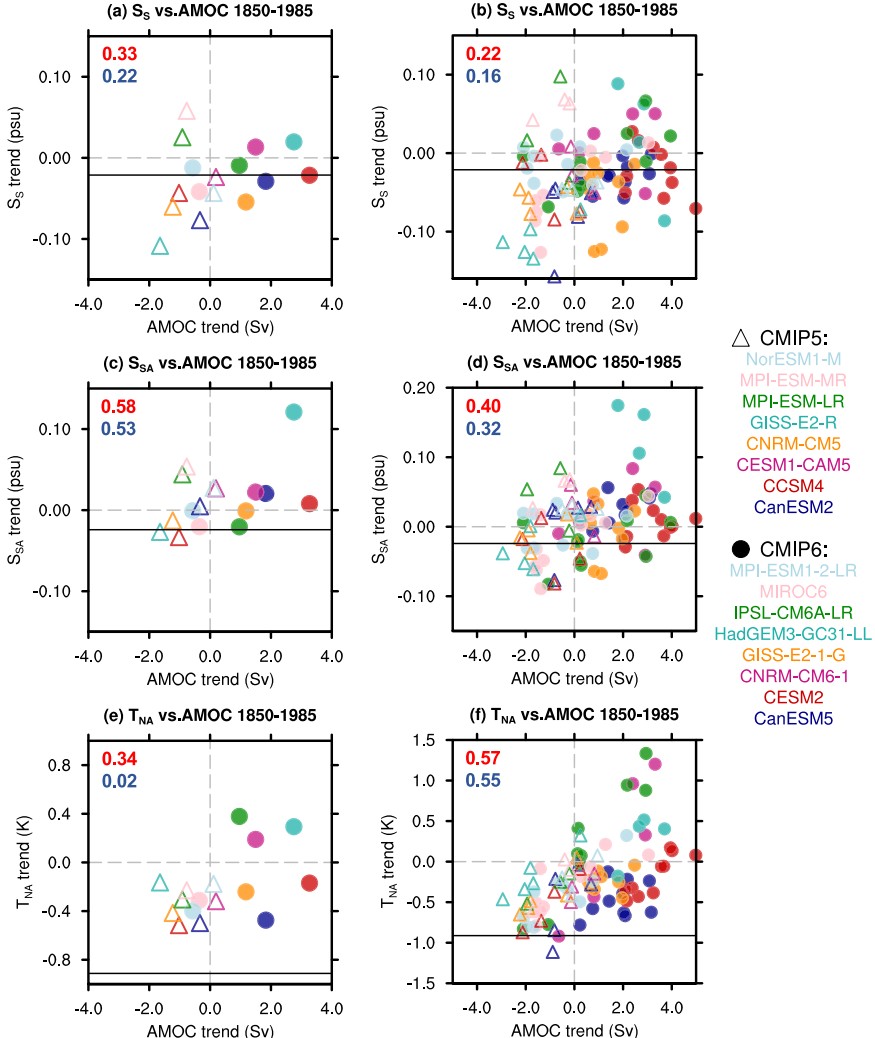

**Fig. 2 | Trend correlation between Atlantic meridional overturning circulation (AMOC) indices for period 1850–1985. a** Scatter for trends of AMOC intensity and its salinity-based fingerprint $S_S$ across model ensemble means. Trends are calculated as the linear change from 1850 to 1985; (**b**) same as (**a**) but across model members. (**c, d**) same as (**a, b**) but for AMOC and the salinity-based fingerprint $S_{SA}$; (**e, f**) same as (**a, b**) but for AMOC and the warming hole fingerprint $T_{NA}$. Solid line in each panel marks the corresponding value in observations (1900–1985 for salinity indices and 1870–1985 for $T_{NA}$). Red and blue numbers in each panel are the correlation coefficients in CMIP6 (dots) and CMIP5 (triangles), respectively.

fingerprint is about twice that of $T_{NA}$ (-0.8 vs -0.4). Results remain robust if the annual $T_{NA}$ is replaced with the cold season $T_{NA}$. This leads to our hypothesis that the SSS fingerprint is an optimal fingerprint that captures the real world AMOC trend better than $T_{NA}$ with a higher signal/noise ratio. The following section describes the multiple lines of evidence and the fundamental dynamics underpinning this hypothesis.

**Trend "Signal" and variability "Noise" in AMOC fingerprints**
Our first modelling evidence for the optimal SSS fingerprint relative to $T_{NA}$ is their trend correlations with AMOC for the period of 1850-1985, which has been used for testing $T_{NA}$[14,27]. In contrast to the reduced trend correlation from cross-member to cross-model between $T_{NA}$ and AMOC (Fig. 2e, f), the trend correlation between the SSS fingerprint and AMOC increases by nearly 50% in both CMIP5 and CMIP6 (Fig. 2b vs a or Fig. 2d vs c). Thus, suppressing internal variability by ensemble mean in each model increases the trend consistency between the SSS fingerprint and AMOC, but decreases the trend consistency between $T_{NA}$ and AMOC. This supports the SSS fingerprint containing more AMOC trend "signal" than $T_{NA}$.

The second evidence of $T_{NA}$ containing more AMOC variability "noise" can be seen in the control simulations, which contains only internal variability under constant external forcing. Figure 3a shows the MMEM of lead-lag correlations between the annual northern AMOC transport (at ~30°N) and its fingerprints $T_{NA}$, $S_S$ and $S_{SA}$ (for individual models see Fig. S3), with the maximum lagged correlations in each model shown in the scatter diagram Fig. 3d for $T_{NA}$ vs $S_S$ (magenta dot) and $T_{NA}$ VS $S_{SA}$ (orchid dot). Both $T_{NA}$ and SSS indices are correlated with AMOC at 95% significance level, confirming the validity of both fingerprints[35]. Relatively, however, $T_{NA}$ has a higher correlation (-0.6) and a short lag of 0-5 years, while the SSS indices have a lower correlation (-0.25) and decadal-scale lag (Fig. 3a, d, Fig. S3), supporting our hypothesis that $T_{NA}$ contains more AMOC internal variability than the SSS fingerprint. The longer lag of SSS indices can be understood as its delayed response to the AMOC change in the South Atlantic, which further lags the AMOC change in the North Atlantic by 5–8 years as seen in the lagged correlation (Fig. 3a, Fig. S3). Similar results can be found in the experiments forced by natural forcing of solar variability and volcanic eruption (Hist-nat), or forced by the stratospheric ozone (Hist-stratO3) (not shown) in the CMIP6 Detection and Attribution Model Intercomparison Project (CMIP6-DAMIP; Table S2; Methods). This is because the variable forcing in these two experiments are relatively weak such that model variability is still dominated by internal variability (green and grey lines in Fig. 4) as in the control simulations.

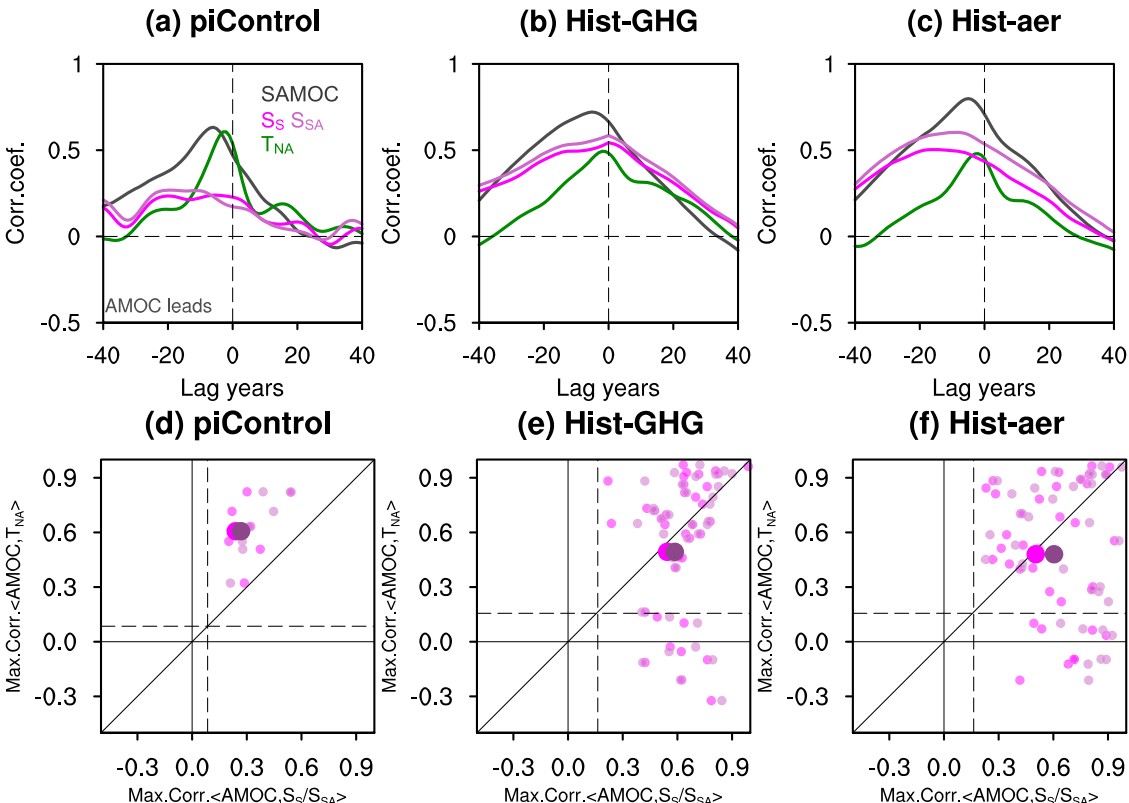

**Fig. 3 | Lead-lag correlation. a–c** Lead-lag correlation between Atlantic meridional overturning circulation (AMOC) intensity and southern AMOC intensity (gray), AMOC and the warming hole fingerprint $T_{NA}$ (green), AMOC and the salinity-based fingerprint $S_S$ (magenta), AMOC and the salinity-based fingerprint $S_{SA}$ (orchid) for piControl (**a**), Hist-GHG (**b**) and Hist-aer (**c**) simulations. Time series is 11 years locally weighted scatterplot smoothing (LOWESS) filtered. **d–f** Scatter diagram of maximum correlations between AMOC fingerprints and AMOC when AMOC leads. Small magenta ($S_S$ versus $T_{NA}$) and orchid ($S_{SA}$ versus $T_{NA}$) dots are for each CMIP6 model member while big ones are for the ensemble mean. Vertical and horizontal dashed lines in (**d–f**) indicate the 95% significance level (determined by Monte Carlo method) for SSS indices (with a lag of 20 years) and for $T_{NA}$ (with a lag of 5 years), respectively. For piControl simulation, the significance level is the averaged value over the seven simulations with different lengths (Table S1).

The third evidence for optimal SSS fingerprint can be seen in the simulations forced by long-term forcing of anthropogenic GHGs (Hist-GHG) and aerosols (Hist-aer) in CMIP6-DAMIP, in comparison with the control simulations. The maximum lagged correlation between AMOC and SSS-based AMOC indices nearly doubles in forced model runs compared with the pre-industrial control run (Fig. 3b, c vs a) and becomes comparable to that for $T_{NA}$. This enhanced correlation is caused by the much-increased AMOC trend signal in the two forced experiments. Indeed, GHGs warming and aerosol cooling are the two leading long-term forcing, which nevertheless have the opposite climate impacts (brown and blue lines in Fig. 1f; Methods). The forced AMOC response exhibits a clear trend, decreasing to GHGs warming and increasing to aerosol cooling[4,20,27] (Fig. 4a). Similarly, the forced SSS indices also show a clear trend response (orange and blue lines in Fig. 4b, c). The seemingly inconsistent SSS indices and AMOC after the 1990s under aerosol forcing is, we speculate, due to the lagged response of the South Atlantic salinity to the AMOC change, as discussed earlier. The SSS response, especially $S_S$, to the GHGs or aerosols forcing is forced predominantly by the AMOC change, instead of the surface E-P forcing associated with the hydrological response[35] (Fig. S4 and S5, Supplementary Text). In comparison, the forced $T_{NA}$ trend response, even using the MMEM, are still distorted substantially by internal variability (orange and blue lines in Fig. 4d). When all the forcing is combined, the competition between the impacts of GHGs and aerosols prior to the 1980s leads to the prevalence of natural AMOC variability[41] (with only weak increasing trend), while the combination of continued increase of GHGs and the reduction of aerosols after the 1980s forces an accelerated AMOC weakening as in the CMIP6

historical runs[27,41] (Fig. 4a, Fig. 1a). This final AMOC response is reasonably simulated by the SSS indices, especially, $S_{SA}$ (Fig. 4b, c), but not by $T_{NA}$. Finally, across simulation members in Hist-GHG and Hist-aer, the maximum lagged correlation with AMOC exhibits strong scatter with some even becoming negative for $T_{NA}$, but has a much smaller scatter with all correlations positive for the SSS indices (Fig. 3e, f). The enhanced lead-lag correlation of SSS indices with AMOC and a small cross-member spread relative to $T_{NA}$ in response to long-term GHGs and aerosol forcing further support our hypothesis that the SSS fingerprint has a higher trend-signal/variability-noise ratio than $T_{NA}$.

**Mechanisms for trend/variability ratio in AMOC fingerprints**
The higher trend/variability ratio in the remote SSS fingerprint than $T_{NA}$ can be understood from the dynamic adjustment of AMOC response to buoyancy forcing over the SPNA. The AMOC response is determined by the basin-wide adjustment of westward propagating long Rossby waves[42]. Since Rossby wave speed increases towards the equator, the basin-wide adjustment is a decaying response[43,44]. In response to variable buoyancy forcing over the SPNA, higher-frequency AMOC variability decays faster southward, especially across the equator[38,45,46], leaving more long-term trend response, or higher trend/variability ratio, remotely in the South Atlantic than the SPNA, and, in turn, in our SSS fingerprint than $T_{NA}$.

This ocean dynamic mechanism is confirmed in our ocean general circulation model (OGCM) experiments forced by idealized heat flux forcing. Figure 5 shows one example, in which the North Atlantic is forced by variable heat flux forcing that consists of a centennial change (180-year period) and interdecadal variability (30 year variability)

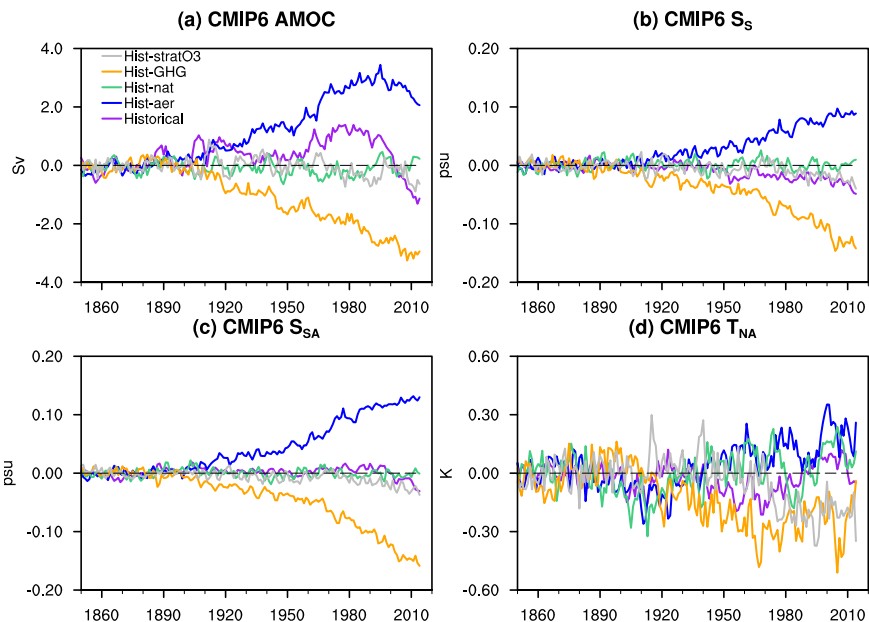

**Fig. 4 | Response of AMOC indices in CMIP6-DAMIP experiments.** Time series of anomalous multi-model ensemble mean (MMEM) Atlantic meridional overturning circulation (AMOC) intensity (**a**), salinity-based fingerprint $S_S$ (**b**), salinity-based fingerprint $S_{SA}$ (**c**) and warming hole fingerprint $T_{NA}$ (**d**) in CMIP6 historical (purple), Hist-aer (blue), Hist-nat (green), Hist-GHG (orange) and Hist-stratO3(gray) simulations. Anomalies are relative to the means of 1850–1900.

(Fig. 5a, Methods). The AMOC interdecadal variability is reduced significantly by ~70% from 30°N to 30°S (Fig. 5c), quantitatively consistent with theoretical studies[45,46]. In comparison, the AMOC centennial trend remains largely unattenuated (reduced by only ~30%) at 30°S (Fig. 5c). As a result, the AMOC trend/variability ratio, as defined as the ratio of standard deviation between centennial and interdecadal variability, increases monotonically southward by more than two times (Fig. 5h). The variation of $T_{NA}$ and Ss resembles that of the AMOC in the North and South Atlantic, respectively (Fig. 5b,c,e–g), as expected. Therefore, the trend/variability ratio of $T_{NA}$ and Ss also resembles that of AMOC in the North and South Atlantic, respectively (Fig. 5h). Finally, the lead-lag correlations of these fingerprints with the AMOC transport also show similar features to those in the forced CMIP6-DAMIP experiments (Fig. 3b, c), with the $T_{NA}$ and Ss of comparable correlations, and with the Ss lagging the $T_{NA}$ and the North Atlantic AMOC by nearly a decade (Fig. 5d).

In the real world, or even coupled general circulation models (CGCMs), the AMOC trend/variability could be complicated by other factors, such as the wind forcing and longer-term internal variability of centennial time scales. Nevertheless, given the stable climate in the STSA trade wind region relative to the storm track region of SPNA, wind forcing seems unlikely to distort the pattern of trend/variability ratio completely. Indeed, this increased trend/variability ratio in the STSA has been found in some CMIP6 experiments, for example, in one member of the historical GHGs forcing experiments in the GISS-E2-1-G model (hereafter GISS-GHGs) (Fig. 5i–l) (other members show similar results, Fig. S6). Similar to our OGCM experiment above, the AMOC in GISS-GHGs is dominated by a 30-yr internal variability along with a weakening trend in response to global warming (Fig. 5i). The trend/variability ratio in GISS-GHGs increases from 30°N to 30°S monotonically by nearly three times for AMOC, and, similarly, from $T_{NA}$ to $S_S$ for the fingerprints (Fig. 5i–l).

Besides the ocean dynamic mechanism above, atmospheric dynamics also favor the salinity fingerprint. $T_{NA}$ is very sensitive to volcanic forcing and stratospheric ozone as seen in the CMIP6-DAMIP experiments Hist-nat and Hist-straO3, where the responses are almost comparable to those to the anthropogenic GHGs and aerosol forcing (Fig. 4d). For example, the forced $T_{NA}$ response in Hist-nat shows two

multi-decadal episodes (1870s–1930s, 1960s–1990s) of minimum (green, Fig. 4d), corresponding to the two observed cooling periods caused by volcanic eruptions[22] (Fig. 1f). Similarly, the forced $T_{NA}$ response to stratospheric ozone variation shows strong multi-decadal variability between the 1980s and 2010s (gray, Fig. 4d), corresponding to the observed multi-decadal variability in spring Arctic stratospheric zone[47]. In comparison, there is little sensitivity to volcanic forcing and stratospheric ozone effect in AMOC (Fig. 4d vs a). This is consistent with studies showing that $T_{NA}$ can be driven by external radiative forcing, or more general, the atmospheric forcing without an explicit role of AMOC change[21,22,48,49]. In contrast to $T_{NA}$, the SSS indices show little response to volcanic and stratospheric ozone forcing (Fig. 4b, c), due partly to the subtropical South Atlantic region being less susceptible to either volcanic eruption[50,51] or Antarctic/Arctic stratospheric ozone variation[47,52,53].

Finally, the stronger sensitivity of $T_{NA}$ than $S_S$ to fast variability forcing can also be seen in CMIP6 experiments of abrupt $CO_2$ quadrupling of 150 years (abrupt-4xCO2, Table S1). In response to the abrupt $CO_2$ increase, $T_{NA}$ exhibits a large opposite (negative) response in initial decades in about half of the models, while $S_S$ and AMOC are both dominated by a slow trend response across all models (Fig. S7). This provides a further support of our hypothesis.

Some studies have also suggested that changes in the South Atlantic could be related to interdecadal variability in the Pacific Ocean through atmospheric teleconnection[54,55] or driven by the Southern Annular Mode (SAM) through its effect on Agulhas leakage[56–59]. Our results suggest, however, a strong control of northern Atlantic buoyancy forcing on AMOC on multi-decadal and longer timescales, with the long-term response emerged most clearly in the South Atlantic, as detected in both the control and forced simulations and supported by theoretical studies[35,45,46].

## Discussion

Given the salinity pileup as the optimal AMOC fingerprint for detecting AMOC weakening in response to current global warming, we interpret the accelerated decline of the SSS index in the 1990s (Fig. 1b, c) as an evidence that an accelerated AMOC weakening forced by the combination of anthropogenic GHGs and aerosol forcing, as predicted by

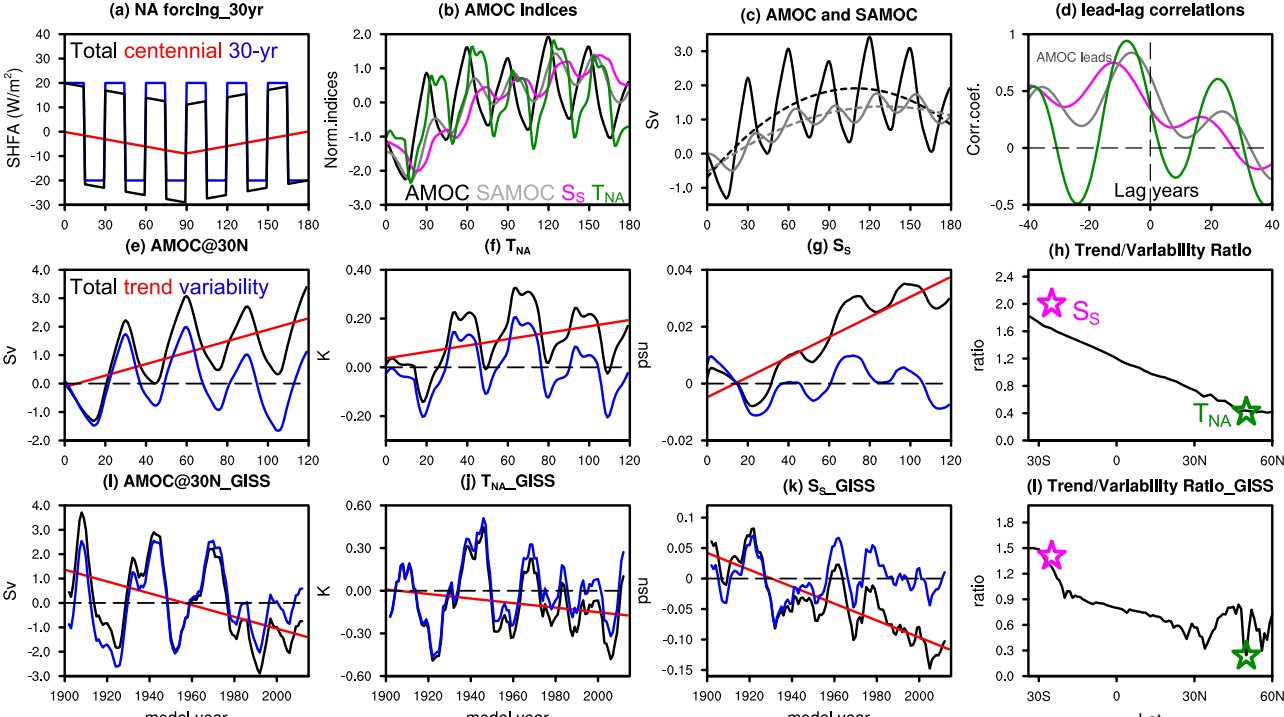

**Fig. 5 | Trend-signal/variability-noise ratio of Atlantic meridional overturning circulation (AMOC) indices. a–h** Demonstrate the results from an ocean general circulation model (OGCM) experiment: **a** Surface heat flux forcing applied in the North Atlantic (black) which consists of two signals: 30-years variability signal (blue) and centennial trend signal (red). **b** Normalized anomaly of the AMOC at 30°N (black), AMOC at 30°S (gray), warming hole fingerprint $T_{NA}$ (green) and salinity-based fingerprint $S_S$ (magenta). Anomalies are relative to the corresponding control experiment. **c** Anomaly of the northern and southern AMOCs. Also

shown are the cubic best-fit regression lines (dashed) for the two. **d** Lead-lag correlation between AMOC (northern AMOC) and southern AMOC (gray), AMOC and $T_{NA}$ (green) and AMOC and $S_S$ (magenta). **e–g** The decomposition of (**e**) northern AMOC, (**f**) $T_{NA}$ and (**g**) $S_S$ into long-term trend (red) and short-term variability (residual; blue). **h** Ratio of standard deviation between centennial and interdecadal AMOC variability as a function of latitude. Magenta and green stars mark the ratio of $S_S$ and $T_{NA}$, respectively. **i–l** Same as (**e–h**) but for a coupled general circulation model (CGCM) simulation (see text).

model experiments (Fig. 1a, Fig. 4a), may have occurred about one decade ago in the 1980s (while an overall weakening in AMOC probably started several decades earlier[13,14]). In the future, the combination of a continued increase of GHGs and decrease of anthropogenic aerosol will likely further accelerate the AMOC weakening. Our study, combined with recent analysis that AMOC may have evolved from relatively stable period to a point close to critical transition[60], may provide early warning signals of accelerated anthropogenic AMOC weakening and its potential climate impact in the coming decades.

## Methods
### CMIP5 and CMIP6 model simulations
We use CMIP6 and CMIP5 experiments to evaluates the AMOC and its indices as well as their response to climate forcing (Table S1). CMIP6 employs the latest generation of climate models. We use the model output of 8 CMIP6 models with a total of 70 ensemble members and 8 CMIP5 models with a total of 31 ensemble members for all-forcing historical simulations. We also use 7 CMIP6 models (with one member for each) for PI control simulations. We found the spread among members within one model is comparable to that among different models. Therefore, the MMEM in the present study is calculated across all the available model members rather than across models' ensemble means, although the results are similar.

We further investigate the attribution of AMOC and its fingerprints using simulations from the Detection and Attribution Model Intercomparison Project (DAMIP; Table S2). As a part of CMIP6, DAMIP isolates the individual effect of anthropogenic GHGs (Hist-GHG), anthropogenic aerosols (Hist-aer), natural forcing (e.g., solar activity and volcanic eruptions; Hist-nat) and stratospheric ozone (e.g., ozone depletion; Hist-stratO3). We define the anthropogenic aerosol forcing

as the total absorbed short-wave radiation (ASR) in the Northern Hemisphere (ASR_NH) derived from the simulations forced by anthropogenic aerosols. Compared with CMIP5, more models with indirect aerosol forcing (e.g., aerosol-cloud microphysical effect) are included in CMIP6. The inclusion of indirect aerosol forcing reinforces the cooling effect of aerosols, leading to an increasing radiative cooling of ~2 W/m² between the 1880s and the 1980s (Fig. 1f). After the 1980s, the cooling effect of anthropogenic aerosols decreases by ~0.5 W/m². Our estimate of aerosol forcing agrees well with the gridded aerosol community datasets (CEDS)[30,61]. The larger magnitude (by ~0.3 W/m²) in our estimate compared with CEDS may be attributed partly to the overestimation of aerosol forcing in CMIP6 models and partly to the climate feedbacks in models, notably the sea ice-albedo feedback (i.e., more reflection of incoming short-wave radiation by increasing sea ice in response to aerosol's cooling effect). The GHGs concentration, on the other hand, increases continually since the industrial revolution, notably a rapid increase in atmospheric $CO_2$ from 290 to more than 400 ppm (Fig. 1f), corresponding to an increase in radiative warming effect of 3.5 W/m² (ref. [61]). We also use abrupt-4xCO2 experiments (Table S1) to study the response time scales of AMOC indices to abrupt $CO_2$ quadrupling.

**Definition of AMOC strength, AMOC fingerprints and AMO index**
AMOC intensity is defined as the maximum overturning streamfunction below 300 m over 30-50° N in the Atlantic. This is also referred to in the text as the northern AMOC. Similarly, the southern AMOC is defined as the maximum overturning circulation below 300 m over 10-34° S in the Atlantic. The "warming hole" SST based index $T_{NA}$, is defined as annual mean, SPNA mean (15–40°W,46–60°N) SST minus annual mean, global mean SST. This definition is a simplified version of

that of ref. [35] but with similar results[27] (not shown). The "salinity pile-up" index $S_S$, is defined as annual mean, STSA mean (averaged over 10–34°S) SSS minus annual mean, STSIP mean SSS at the same latitude band. The MMEMs for these two indices are calculated with each ensemble member equally weighted. The AMO index shown in Fig. 1e is calculated after ref. [14] as the weighted mean SST over the North Atlantic (0° N to 60° N) relative to the mean SST from the period 1900-1950 with the global mean SST (60° S to 60° N) removed.

### Ocean model and sensitivity experiments

The ocean model used in our study is POP2[62,63], which is the ocean-component of the coupled CESM model. Our version of POP2 has a uniform resolution of 3.6° in the zonal direction and a non-uniform resolution (0.6° near the equator, gradually increasing to the maximum of 3.4° at 35° N/S and then decreasing poleward) in the meridional direction. The model has 60 levels in the vertical, with a uniform resolution of 10 m in the upper 160 m, increasing to 250 m to the depth of 3500 m and then remaining the same towards depth. The control run is forced by the normal year forcing from the Co-ordinated Ocean–Ice Reference Experiments (CORE) dataset[64], using the CORE experimental design as outlined in ref. [65]. The CORE forcing and bulk formulas used here are the version 2 (COREv2) as defined in ref. [64]. This model has been used for the demonstration of the mechanism of salinity pileup fingerprint[35].

The CTRL run is integrated for 900 years when AMOC indices are found near quasi-equilibrium, from which the sensitivity experiment is launched. Sensitivity experiment is integrated 180 years. Anomalous surface heat flux is imposed to produce AMOC variability with different time scales. The magnitude of 30-year periodic forcing is 20 W/m². Besides the periodic forcing, we also apply a linear heat flux anomaly over the North Atlantic (20°N–80°N), with its magnitude decreasing linearly from zero to -9 W/m² in 90 years and then increasing linearly to 9 W/m² from year 91 to180. The reversal of linear forcing causes a trend reversal in AMOC strength, resembling that observed in CMIP6. The wind stress remains unchanged in our sensitivity experiments so that there is no effect of wind induced circulation change. As such, the dynamic effect on salinity transport is caused by the buoyance-forced AMOC change only.

### Data availability

CMIP6 outputs including DAMIP and abrupt-4xCO2 outputs are publicly available at https://esgf-index1.ceda.ac.uk/projects/cmip6-ceda/. CMIP5 outputs are publicly available at http://www.ipcc-data.org/sim/gcm_monthly/AR5/Reference-Archive.html. Ocean salinity data: (1) Hadley Centre EN4 dataset version 4.2.2 with the Gouretski and Reseghetti bias corrections applied[39,66] at https://www.metoffice.gov.uk/hadobs/en4/download-en4-2-2.html. (2) ISHII data[40] version 6.13 at https://rda.ucar.edu/datasets/ds285.3/; Ocean temperature data: (1) HadISST[67] from Hadley Centre, https://www.metoffice.gov.uk/hadobs/hadisst/; (2) ERSSTv5[68] at https://psl.noaa.gov/data/gridded/data.noaa.ersst.v5.html. RAPID array along 26.5° N is publicly available on www.rapid.ac.uk/rapidmoc/rapid_data/. The South Atlantic AMOC derived from Argo and altimetry are freely available from http://www.aoml.noaa.gov/phod/samoc_argo_altimetry/index.php. Stratospheric aerosol optical depth and atmospheric $CO_2$ measurements are available from NASA GISS (http://data.giss.nasa.gov/modelforce/strataer/ and https://data.giss.nasa.gov/modelforce/ghgases/). The ECCO's latest ocean state estimate, ECCO Version 4 release 4 (V4r4), covering the period 1992–2017, is freely available from https://www.ecco-group.org/products-ECCO-V4r4.htm. The GloSea5 reanalysis data is available at https://resources.marine.copernicus.eu/product-detail/GLOBAL_REANALYSIS_PHY_001_031/INFORMATION. In addition, the POP2 data of sensitivity experiments used in this study is available at https://doi.org/10.5281/zenodo.7534271 or from the corresponding authors upon request.

### Code availability

POP2 is freely available as open-source code from http://www.cesm.ucar.edu/models/cesm1.1. The code used to analyze and visualize the data is written with NCL 6.6.2 (https://www.ncl.ucar.edu/). The scripts for main figures are available at https://doi.org/10.5281/zenodo.7534271 or from the corresponding authors upon request.

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

## Acknowledgements

We acknowledge the computing support from Laoshan Laboratory, the National Key Scientific and Technological Infrastructure project "Earth System Science Numerical Simulator Facility" (EarthLab) and NCAR's Computational and Information Systems Laboratory (CISL). This work is jointly supported by the National Natural Science Foundation of China (42106013 to C.Z., 41830964 to S.Z.), Science and Technology Innovation Project of Laoshan Laboratory (LSKJ202203300 to C.Z., LSKJ202202200 to S.Z.), the US Department of Energy (CW33566 to Z.L.), National Oceanic and Atmospheric Administration (NA20OAR4310403 to Z.L.) and Shandong Province's "Taishan" Scientist Program (ts201712017 to S.Z.).

## Author contributions

Z.L. and C.Z. conceived the study and wrote the paper. C.Z. performed the analyses and experiments. S.Z. and L.W. contributed to the discussion of the results.

## Competing interests

The authors declare no competing interests.
