## [Peer Review File · Nature Communications]

Likely Accelerated Weakening of Atlantic Overturning Circulation Emerges in Optimal Salinity FingerprintREVIEWER COMMENTS

Reviewer #1 (Remarks to the Author):

General remarks: This paper argues that a SSS based AMOC index from the STSA (subtropical South Atlantic) has a better signal-to-noise ratio than AMOC indices from the North Atlantic (especially the SST-based index). This is tested in model simulations. From the observed SSS index they conclude that the weakening of the AMOC has accelerated since the 1980s.

The paper is very well written and easy to understand, and it is certainly a very interesting topic. But while I think that the analysis of the different indices with regards to their signal-to-noise ratio is nicely done, and the arguments for using the SSS indices for a clearer long-term signal are convincing, I don't see enough support for the claim of an accelerated AMOC weakening since the 1980s. Also, I think that a more in-depth study of the actual time lag between AMOC and SSS index as well as a better understanding of the time scales at which AMOC changes are represented in the SSS index (decadal, multi-decadal) is needed. I therefore suggest major revisions of the paper.

My problems with the conclusions have several reasons:

1. It is assumed that the MMEM correctly simulates the forced response of the AMOC (e.g., as it is used to argue that the discrepancy between observed T_{NA} and model forced T_{NA} is due to "noise"/internal variability), but due to the mentioned possible overestimation of the aerosol effect in CMIP6 this does not have to be true, i.e., it could be that the forced response in the models does not agree with the forced response in the real world.
2. Even though the SSS index has a better signal-to-noise ratio, it still appears to show a smaller absolute trend than the T_{NA} or the actual AMOC (figure 5c). This raises the question of how it is possible that the SSS index shows an accelerated weakening of the AMOC since the 80s while other reconstructions don't show a trend at all (e.g. Worthington et al. 2021 or Fu et al. 2020)?
3. Given the large lag times that you see for the SSS index (figure 3a) – how do you know that the signal in the SSS index post 80s does not correspond to the post 50 weakening found in the T_{NA} index?
4. How is the SSS index better than a low frequency filtered T_{NA} index (which would also reduce the noise?) Just because the signal-to-noise ratio in the SSS index is better, this does not mean that it is a better representation of the actual AMOC (as the AMOC might display large internal variability).
5. Why is there this discrepancy between observed SSS indices and modelled SSS indices (figure 1b,c). If the SSS index mainly picks up the forced AMOC signal, shouldn't it be very similar to the MMEM SSS indices?

General remarks:

When you talk about AMOC strength in the models: why do you use this more flexible definition instead of looking at 26N or 30N AMOC to make it better comparable to the observations? The same question goes for the South Atlantic AMOC, looking at 34.5S to make it comparable to SAMBA might be nice.

For your definition of T_{NA} you use annual data (instead of Nov-May SST) and a larger region than ref.34 – this of course could be one of the reasons why you increase the noise in the index, as your paper concentrates on finding the best signal-to-noise ratio, this is rather unfortunate. While I understand that one could argue for a larger region, especially when looking at models as the region most sensitive to AMOC heat transport changes might differ from model to model, I strongly suggest also only using Nov-May or at least the cold season SST to minimize the more chaotic influence of the atmosphere on the index.

Further comments:

ll. 64ff/Figure 1d) This shows the CMIP5 and CMIP6 T_{NA} is very different from the observed one, as the former do not show a long-term weakening over the 20th century. I assume MMEM is multi-model ensemble mean? This should be defined.

ll.69-70 "strong strengthening" sounds strange, maybe rephrase

ll.70-71 Figure 1d does not show the AMV, if you refer to the AMV here it would be good to show, also what do you mean when you say the T_{NA} is overwhelmed by the AMV, that it drives most of the observed signal? As AMV likely contains an AMOC signal this an influence of AMV on T_{NA} does not exclude the AMOC as the main T_{NA} driver.

- Il. 90 ff “implying a forced AMOC response likely of little to slight weakening trend before 1980s” – this is speculative maybe rephrase this?
- Il. 93 ff Can you be more specific about the trends in CMIP5/6 after and before 1980 (i.e. is this the linear trend over 40 years? If so, it would be great to compare it to the 1920-1980 trends).
- Il. 96 ff/Figure 1a I am a little surprised about the GloSea5 data (purple line), as the new Jackson et al., 2022 paper shows that basically all AMOC reconstructions hint at a strengthening of the AMOC from the 80s to late 90s (including the GloSea5 data) – can you explain this discrepancy?
- Il.114ff If the indices behave similarly, does this means S_STSIP is approximately constant or just very small? Please comment. Plus, in figure 2 the results/correlation coefficients are actually quite different for the two indices, not supporting that they behave very similar.
- Il.117ff I am not sure whether I agree: Why do you assume that the remote response can only be seen in your fingerprint?
- Il.120 ff Could you give the correlation between AMOC and SSS index for the different models (both cross-member and cross-model) to support this?
- Il. 135 ff While it is true that there seems to be more forced “signal” in the SSS fingerprints compared to the T_NA the actual correlation between AMOC and index is for individual members higher for T_NA. This indicates that T_NA is a better representator of the actual AMOC (which of course can contain internal variability).
- Il.150 ff Why does a higher correlation and a smaller lead time correspond to more noise/internal variability?
- Il. 164 ff Why does the lag between AMOC and SSS index change for the different simulations (from about 20 years in piControl to only a ~1-2 years in Hist-GHG)? What kind of mechanism could cause this, and don't you think that this big change could be indicating that the SSS index does not work that well??
- Il. 174 ff While you convincingly show that changes E-P hardly affect the SSS indices, this does not mean that the changes have to be due to AMOC changes. They could also relate to changes in the gyre circulation or other local ocean circulation changes – how do you know that that is not the case?
- Il. 181 ff It seems to me that the SSS indices cannot capture this final response to GHG and aerosols consisting of an increase in AMOC strength in the 80s and following decrease, either – even though the signal has a fairly high amplitude. Did you expect this?
- Il. 191 ff You are citing a study about a 2-layer model, can you provide evidence that the real world AMOC response is set by Rossby waves and give an estimate of the expected time lags?
- Il. 244 ff Why is this accelerated slowdown that is, as you state, predicted by CMIP5/6 not visible in the models' SSS indices (Figure 1b,c)? That does not make sense.
- Figure 5/S6 Can you comment on why the T/V ratio in all models seem to show a minimum right at the location/latitude of the T_NA index? This made me curious.

References

- Worthington, E. L., Moat, B. I., Smeed, D. A., Mecking, J. V., Marsh, R., and McCarthy, G. D.: A 30-year reconstruction of the Atlantic meridional overturning circulation shows no decline, *Ocean Sci.*, 17, 285–299, <https://doi.org/10.5194/os-17-285-2021>, 2021.
- Fu et al., 2020: <https://doi.org/10.1126/sciadv.abc7836>
- Jackson et al., 2022: <https://www.nature.com/articles/s43017-022-00263-2>

Reviewer #2 (Remarks to the Author):

Review of “Accelerated Slowdown of Atlantic Circulation Emerged in Optimal Fingerprint” by Chenyu Zhu and colleagues.

This manuscript addresses the question, “Has AMOC slowed down over the last century”? The topic is timely, controversial and of considerable scientific interest. The authors are building off of the work by two of the present authors (<https://doi.org/10.1038/s41558-020-0897-7>) where they identify a “fingerprint” of AMOC variability in the salinity of the South Atlantic, specifically the difference in salinity between the tropical to subtropical South Atlantic (10S-34S) and the Indo-Pacific (10S-34S). In the current work, they apply that fingerprint using EN4 salinity data and argue, based on several lines of evidence that the southern hemisphere salinity proxy is better than a North Atlantic “warming

hole" index for monitoring decadal to centennial-scale trends in AMOC.

I think the salinity index idea is exciting, novel, and has promise for getting us closer to understanding the temporal development of AMOC over the last century. The idea is a substantial leap forward for which the authors should be congratulated and deserves to be published. I am recommending major revisions because of three factors. First, I have deep reservations about the use of the EN4 salinity data for the analysis of salinity trends, given that salinity observational datasets tend to be sparse in both the South Atlantic and tropical South Pacific. Since these data form the primary results referred to in the title, it is important that they be robust. However, I also note that a great deal of the work described by this manuscript involves using model data to better understand why the salinity data gives a better perspective on the multidecadal trend of AMOC than does a North Atlantic sub-polar gyre-based index. If the EN4 data turn out to be unusable in the early period of the analysis, I think it would be very reasonable to change the focus of the paper to an analysis of why the salinity proxy of AMOC captures multidecadal trends in AMOC better than North Atlantic temperature proxies of AMOC.

The second aspect of the current manuscript that gives me pause is that the text never really explains how the salinity build-up is caused by AMOC slow-down. I understand the current work is building on the prior paper, but even reading the prior paper, I was disappointed with the lack of a mechanistic explanation in either paper. What causes the salinity convergence or divergence in the southern hemisphere surface? I would expect some explanation involving salty Agulhas Leakage, northward low salinity water crossing the equator, and air-sea freshwater fluxes, but the current manuscript lacks even the briefest of explanations along similar mechanistic lines. Since this is the basis for the paper, I think it is worth addressing.

The third factor that must be addressed before publication is the writing itself. The writing is often very unclear because of the lack of precision in the use of words and sometimes the incorrect usage of words. Often statements are ambiguous as written. I could guess from the context what the authors intended, but this needs to be corrected before publication. Additionally, the manuscript is littered with grammar errors, which can be relatively easily corrected. Both of these things combine to make the manuscript very difficult to read and comprehend. I give only examples in the specific comments because it became rather tedious to point out every grammar mistake or ambiguous statement and my job is to review the science, not copy edit the manuscript.

Some specific comments for improving the manuscript are listed below.

SPECIFIC COMMENTS

More on Trends in EN4 data and data coverage.

It is unclear to me if the authors are using the "raw" data or the infilled and gridded "analysis" data of EN4. I assume the gridded analysis field. EN4 relaxes to a 1970-2000 climatology when data are sparse. Given that salinity observations tend to be sparse, I think it is vital for any interpretation of long-term trends to check the observation weights in the data. It is worth a line on the main figure (figure 1) that demonstrates that the data are based on some measurements and not just relaxed to climatology (for example, plotting the number of observations in the region, or % of grid boxes where the weight is majority on the data rather than climatology).

I say this because the EN4 Product Users Guide specifically cautions against analysis of trends in data sparse areas. Relevant text is reproduced from the guide available at the link below:

https://www.metoffice.gov.uk/hadobs/en4/EN.4.2.2_Product_User_Guide_v1.0.pdf

"6.5 CAN I USE EN4 FOR TREND ANALYSIS?"

EN4 is an observation based [sic] data set and therefore coverage varies in both time and space. For the complete fields provided in the analyses we strongly encourage users to look at the 'observation weights' variables. These will inform users how much the analysis value has been influenced by observations (values closer to one) and how much it has been determined by background fields (values closer to zero). We would not encourage the use of EN4 analyses for trend analysis in areas

where the observation weights are low.”

Line 43: “the collaborative forcing” should probably read “combined forcing” since “collaborative” indicates people working together and is not used for inanimate forces.

Line 49 “continually collaborative anthropogenic forcing” see above for proper use of “collaborative”.

Line 59-60. The text and references imply that the high latitude North Atlantic is the only target for AMOC reconstructions, but Rong Zhang developed a “fingerprint” for AMOC in the equatorial Atlantic. I think just adding in this reference would be enough to address the issue, because technically, the low-latitude fingerprint is also in the North Atlantic.

Zhang R (2007) Anticorrelated multidecadal variations between surface and subsurface tropical North Atlantic. *Geophys Res Lett* 34:1–6.

Line 68 “An application of TNA to real world has been” is one of many examples where the word “the” is missing. It should read “An application of TNA to the real world has been”

Line 86 “Before 1980s” should be “Before the 1980s”. This is consistently in error throughout the paper.

Line 89 “is highly uncertainty” should read “is highly uncertain”

Section starting on Line 100, “Optimal Fingerprint...” This section is where I think the reader needs a more mechanistic explanation. Since this is the foundation of the current work, I think it is perhaps worth a figure, but certainly a few sentences to explain it better. I have to admit the explanation in the author’s previous paper sounded to me more like an assertion rather than a reasoned explanation and the “mechanism schematic” buried in the supplementary text was overly simplistic to the point of not being very helpful. I caution the authors to not repeat earlier mistakes.

Line 115: As a reader, I’m not convinced at this point in the text that $S_s = S_{s_a}$ as asserted here. Because of this, the associated parenthetical phrase sounds more like sloppiness than rigorous science. I think it is worth being specific about which (S_s or S_{s_a}) you are using throughout the paper because although by the end of the paper, I would agree with you, the reader has no reason to believe the authors at this point in the text.

Line 119-120 “SPNA deep-convection region, which is known to force AMOC signal propagating southward coherently” While I don’t argue that deep convection creates a “push” for AMOC, I see no reason that the trend or variability couldn’t be also caused by the pull of upwelling/diapycnal mixing (see Visbeck, M. (2007), Power of pull, *Nature*, 447(7143), 383, doi:10.1038/447383a. and two recent papers by Robbie Toggweiler <https://doi.org/10.1029/2018JC014794> and <https://doi.org/10.1029/2018JC014795>).

Line 132: Sentence “This hypothesis is supported by multiple evidences and, more importantly, is underpinned by fundamental dynamics, as discussed below.” As the reader reads the first part of this sentence, they may expect this to be a summary of the above section which has not brought forth the multiple evidences. It is not until the end of the sentence that the reader learns this sentence is transitioning to the next section. Please change to something like “ The following section describes the multiple lines of evidence and the fundamental dynamics underpinning this hypothesis.

Line 136-137 this sentence is referring to model data and should describe it as such.

Lines 164-166 This sentence is a great example of the lack of clear communication in the writing that makes this manuscript frustrating to read. “The lead-lag correlation with AMOC is more than doubled for the SSS fingerprint (Fig.3 b, c vs a), such that the correlation for SSS index becomes comparable or even greater than that for TNA.” The ideas I think the authors are trying to communicate are more

effectively stated in this sentence," The maximum lagged correlation between AMOC and SSS-based AMOC indices nearly doubles in forced model runs compared with the pre-industrial control run and becomes comparable to that for TNA." Note that the two data sets being correlated (AMOC and SSS fingerprint) are specified and the source of the data, which is relevant for understanding the result, is also specified. Lack of such specificity in the writing generally makes this paper more difficult to read than necessary.

Lines 174-176. "The SSS response to GHGs or aerosols forcing, especially $S!$, is forced completely by the AMOC change, instead of surface E-P forcing associated with the hydrology response³⁴". This is a very strong assertion with little evidence and no stated mechanism. I understand that you are citing the prior work, but all I got out of reading reference 34 were some vague comparisons between the patterns of E-P trends vs the patterns of SSS trends and some more assertions that it is AMOC-related. This is a key concept in your paper and should not be treated so lightly.

Line 177. "even after MMEM" grammar issue. Should read something like, "even using the MMEM"

Line 178-179 "When all the forcing combined, the competition of GHGs and aerosols impacts prior to 1980s lead to a prevalence of natural AMOC variability". The sentence has multiple grammar issues. It should read something like, "When all the forcing is combined, the competition between the impacts of GHGs and aerosols prior to the 1980s leads to the prevalence of natural AMOC variability."

Line 190 "trend-signal/variability-noise ratio" At this point in the manuscript the reader is familiar with the fact that the authors consider the trend to be the signal and the variability to be the noise. This can be simplified here and below as trend/variability ratio so that the text remains specific.

Line 191 "to climate forcing." Please be specific about what climate forcing. I think the authors mean to indicate radiative forcing, but there are other climatic forcing components of AMOC (e.g., winds and freshwater fluxes). Again, this is an example of the imprecise use of language.

Line 191-193. This blithely assumes the "push" is more important than upwelling/diapycnal mixing "pull" in AMOC adjustment. I'm not sure this is correct. I think it is worth a sentence justifying this perspective. See Visbeck, M. Power of pull. *Nature* 447, 383 (2007). <https://doi.org/10.1038/447383a> and two papers by Toggweiler et al 2019 10.1029/2018JC014794 and 0.1029/2018JC014795.

Line 250 I caution against the use of "emerge" because it indicates "climate emergence" which has a specific meaning not intended here.

Line 445 "Our estimate of aerosol forcing is in good agreement with the gridded aerosol community datasets (CEDs) with $\sim 0.3\text{W/m}^2$ stronger, which..." The "with $\sim 0.3\text{W/m}^2$ stronger" is awkward at best and grammatically wrong at worst. The sentence is long and might be better if split into two sentences.

Line 557 I am concerned by defining AMOC without the surface layers when much of the return flow to the North Atlantic occurs in the surface layers. It is my understanding that in the Atlantic, the Subtropical Cells are integral to the overall MOC.

Reviewer #3 (Remarks to the Author):

This is my review of the paper entitled "Accelerated Slowdown of Atlantic Circulation Emerged in Optimal Fingerprint", submitted to

The paper relies on the assumption that the surface salinity in the South Atlantic can serve as a proxy for the AMOC changes, in that a decrease in the AMOC causes by anomalies in the subpolar North Atlantic would reflect in salinification of the South Atlantic. An index is then created, and compared

against CMIP5/6 class models under different forcing scenarios. The paper states that this fingerprint has a better representation of the global warming signal than the previously used subpolar index.

The paper is well written, and makes an interesting/valid point. Salinity in the South Atlantic, for being an integrated variable, and less affected by surface fluxes and direct aerosol forcing, is a potential good fingerprint for ocean circulation. Salinity unfortunately suffers from lack of early data, and model biases. In addition, the South Atlantic does not only respond to the North Atlantic, but also has its own variability, since it is the link of three ocean basins. My comments go in this direction, and additional analysis/discussion is needed to guarantee the feasibility of this index under uncertainties and regional forcings.

Main comments:

1) It would be more informative to define an equation ($amoc \sim a \cdot sal + B$, for example) for this relationship. Fig 1 was too qualitative, and we cannot really see this relationship there. This has been defined for the TNA index (Caesar et al., 2018). Correlations can be OK, but they do not show a clear view of the relationship of the magnitude of changes. I wonder if we can draw a unique relationship between the two indices.

2) Trends in reanalyses and AMOC reconstructions:

I am a bit concerned about the discussion in lines 96-99. The authors claim that "this post-1980s AMOC weakening trend appears consistent with the AMOC reconstructions based on instrument measurements after ~1990s". First, a very small number of salinity observations were available in the South Atlantic until 2007, so most reanalyses suffer from the same bias from the Argo spinup since 2005. In addition, ref 33 state that: 1) the regions between 20 and 25S is dominated by the subtropical gyre, so the changes observed in that region may be just an expression of the wind stress curl changes in the region; 2) There is actually an increase trend of AMOC/MHT in 35S (Figure 14 in Ref 33), so I wonder why the present paper does not include 35S in the average estimate.

3) Teleconnections to SA:

The South Atlantic is a place under influences of atmospheric teleconnections from the Pacific Ocean. Of particular importance here is the PDO, which has influences on long timescales in the South Atlantic (Lopez et al., 2016; Majumder et al., 2019, among others). So I wonder if part of the signal could be driven by teleconnections from the Pacific.

4) The Southern Annular Mode (SAM) effect on the Agulhas leakage and SA salinity has been explored in many papers (Marini et al., 2011, Goes et al., 2014, Durgadoo et al., 2013, Loveday et al., 2015, etc.), in which the southern hemisphere westerlies would force the changes in salinity, and later on potentially influencing the AMOC. SAM has suffered strong trends over the historical period. I wonder if this is the signal driving the changes in the SA. Indeed, these changes would bring SS anomalies in the same signal as the anomalies described in this paper, but via Agulhas Leakage high salinity input from the Indian Ocean.

5) This is just a curiosity, since the authors have experience in this subject. According with the salt advection theory, does the model AMOC stability state influences if the South Atlantic gets saltier or fresher with the AMOC reduction? I asked this question following this statement in L.252 "Our study, combined with recent analysis that AMOC may have evolved from relatively stable period to a point close to critical transition"

Other comments:

Fig 1a- why are the timeseries stopping in 2015? Most of these indices could be extended further on to the present.

Fig 3 - It is really hard to see the difference between red and orange dots.

Green and black correlations (TNA and SAMOC respectively) are much more stable than red and

orange.

L.89 -Typo: Highly "uncertain"

L. 115-117 The two indices should indeed behave similarly because salinity is more or less conserved globally, different than temperature.

L.120 "In contrast to all previous AMOC fingerprints, our SSS index is unique in that it represents the remote AMOC response to the buoyancy flux in the SPNA deep-convection region, which is known to force AMOC signal propagating southward coherently (35-37)"

I do not know how the authors arrived in this conclusion. It is not clear to me what forces what. Is SSS responding to the AMOC weakening or forcing it? It would be good to show some meridional hovmollers of salinity to define this propagation.

L.112-125 "For the real world, the observed S_s or S_{SA} shows a slow decreasing trend in EN4 data prior to 1990s (albeit uncertain in the Ishii data, which is too short), but a clearly accelerated weakening after 1990s in both EN4 and Ishii data sets (Fig.1b,c, Fig.S2a)." How significant is this? There is very little salinity data in the South Atlantic previous to 2005 (See main comment 2).I would suggest the authors to look at the "coverage weights", which is provided by EN4 dataset to infer some significance of the data trends, particularly prior to 1990.

L. 174-176 The statement that " S_s is forced completely by the AMOC change, instead of surface E-P forcing associated to the hydrology response" is not shown correctly in Figures S4 and S5. E-P should be associated to salinity changes (dS/dt) not a direct relationship as shown in Figure S4. To analyze it correctly, S_s should be compared to the time integral of E-P ($dS \sim (E-P) \cdot dt$).

Figure S3: As far as these simulations go, the NAMOC preceds the SAMOC in most simulations. The correlations with S_{SA} are in general small, although according to the authors they are significant. It is really strange that correlations of ~ 0.05 are statistically significant. Am I missing something? It deserve explanation.

References:

Marini, C., Frankignoul, C., & Mignot, J. (2011). Links between the Southern Annular Mode and the Atlantic Meridional Overturning Circulation in a Climate Model, *Journal of Climate*, 24(3), 624-640.

Goes, M., Wainer, I., and Signorelli, N. (2014), Investigation of the causes of historical changes in the subsurface salinity minimum of the South Atlantic, *J. Geophys. Res. Oceans*, 119, 5654– 5675, doi:10.1002/2014JC009812.

Durgadoo, J. V., B. R. Loveday, C. J. C. Reason, P. Penven, and A. Biastoch (2013), Agulhas leakage predominantly responds to the Southern Hemisphere westerlies, *J. Phys. Oceanogr.*, 43, 2113– 2131.

Loveday, B. R., Penven, P., and Reason, C. J. C. (2015), Southern Annular Mode and westerly-wind-driven changes in Indian-Atlantic exchange mechanisms. *Geophys. Res. Lett.*, 42, 4912– 4921. doi: 10.1002/2015GL064256.

Lopez, H., Dong, S., Lee, S.-K., & Campos, E. (2016). Remote influence of interdecadal Pacific oscillation on the South Atlantic meridional overturning circulation variability. *Geophysical Research Letters*, 43, 8250– 8258. <https://doi.org/10.1002/2016GL069067>

Majumder, S., Goes, M., Polito, P. S., Lumpkin, R., Schmid, C., & Lopez, H. (2019). Propagating modes of variability and their impact on the western boundary current in the South Atlantic. *Journal of Geophysical Research: Oceans*, 124, 3168– 3185. <https://doi.org/10.1029/2018JC014812>

Response to Reviews

Reviewer #1 (Remarks to the Author):

General remarks: This paper argues that a SSS based AMOC index from the STSA (subtropical South Atlantic) has a better signal-to-noise ratio than AMOC indices from the North Atlantic (especially the SST-based index). This is tested in model simulations. From the observed SSS index they conclude that the weakening of the AMOC has accelerated since the 1980s.

The paper is very well written and easy to understand, and it is certainly a very interesting topic. But while I think that the analysis of the different indices with regards to their signal-to-noise ratio is nicely done, and the arguments for using the SSS indices for a clearer long-term signal are convincing, I don't see enough support for the claim of an accelerated AMOC weakening since the 1980s. Also, I think that a more in-depth study of the actual time lag between AMOC and SSS index as well as a better understanding of the time scales at which AMOC changes are represented in the SSS index (decadal, multi-decadal) is needed. I therefore suggest major revisions of the paper.

Response to the overall comment: We appreciate the reviewer's constructive comments. Below are our point-by-point responses to the reviewer's questions. We copied the original review in blue and write our reply in black. Changes in the manuscript text are shown with red color highlighting.

My problems with the conclusions have several reasons:

1. It is assumed that the MMEM correctly simulates the forced response of the AMOC (e.g., as it is used to argue that the discrepancy between observed T_{NA} and model forced T_{NA} is due to "noise"/internal variability), but due to the mentioned possible overestimation of the aerosol effect in CMIP6 this does not have to be true, i.e., it could be that the forced response in the models does not agree with the forces response in the real world.

R1.1: We agree with the possibility of model problems. Therefore, we changed the sentence related to this point on by adding a "likely" acceleration in observations. *"In contrast, after the 1980s, the GHGs continued increasing while the aerosol forcing started decreasing⁴³⁰ (Fig. 1e), forcing an accelerated global sea surface warming at a rate of ~0.8°C/50years in model*

simulations (~0.1°C/50year in 1920-1980; not shown) and likely also in observations³⁻⁴.” We have also added “likely” in front of acceleration in many places in the paper, including the title. Nevertheless, based on physical reasoning and previous model studies, we believe this acceleration is very likely. It is known that the aerosol effect bears large uncertainty in models, notably a possible overestimation in CMIP6. Physically, in both observations and models, the competing effect is present between rising CO₂ and rising aerosols before the 1980s. This competing does have a large uncertainty mainly due to aerosols, leading to some uncertainty in AMOC change across models (weakening in CMIP5 and strengthening in CMIP6). After the 1980s, however, CO₂ rising and aerosol decreasing consistently favor a weakening AMOC in both CMIP5 and CMIP6 and resembles the AMOC change during the overlapping period in ECHO reanalysis (Figure. 1a). Therefore, we focus mainly on the post-1980s change and suggest that the combined effects of the two forcing likely trigger an accelerated weakening of the AMOC. This acceleration is certainly true in models, but due to the same physics, we think it is also likely true in observations.

2. Even though the SSS index has a better signal-to-noise ratio, it still appears to show a smaller absolute trend than the T_{NA} or the actual AMOC (figure 5c). This raises the question of how is it possible that the SSS index shows a(n accelerated) weakening of the AMOC since the 80s while other reconstructions don't show a trend at all (e.g. Worthington et al. 2021 or Fu et al. 2020)?

R1.2: Good question. There are large uncertainties in AMOC reconstructions due to sparsely sampled sections and too short time coverage. As the reviewer mentioned, some studies show a stable AMOC between 1980s and 2010s (e.g., Worthington et al.,2021; Fu et al., 2020), different from other reconstructions, including reanalysis products (e.g. composed in our new Fig.S1). Although we can't reconcile the discrepancies among those different reconstructions, partly, we think this discrepancy may be caused by the strong decadal-to-multidecadal variability and the low signal(trend)-to-noise(variability) ratio in the North Atlantic, especially in the SPNA, for the past three decades. As an example, Piecuch (2020) reconstructed Florida Current transport using sea-level records for the past century. The reconstruction suggested weakening trend in deep return flow during longer period starting before the 1950s. However, the trends become insignificant for shorter periods beginning more recently after the 1950s . This

example suggests that any credible detection of long-term trend should be made with a sufficiently long time period. In our study, we based our argument on the longer SSS records and found an unprecedented salinification in the South Atlantic (relative to South Indo-Pacific) since the 1990s. Given the time lag between our SSS index and subpolar AMOC of decadal timescale, we speculate that an accelerated weakening of AMOC has likely occurred since the 1980s. This is supported by the forced response in model simulations. Nevertheless, the point of this review is well taken. Given the data uncertainty (please see our reply in R2.1), as in Piecuch (2020), we have softened our statement on the acceleration by adding “likely” in the title and many other places in the text.

Reference:

Piecuch, C.G. Likely weakening of the Florida Current during the past century revealed by sea-level observations. *Nat Commun* **11**, 3973 (2020). <https://doi.org/10.1038/s41467-020-17761-w>.

3. Given the large lag times that you see for the SSS index (figure 3a) – how do you know that the signal in the SSS index post 80s does not correspond to the post 50 weakening found in the T_NA index?

R1.3: Great question. Given the large lag times for the SSS index, one can speculate that the AMOC weakening has occurred several decades earlier. The post-50s weakening in T_NA index is unique and most likely a mixture of anthropogenic signal (negative AMOC trend related to global warming) and internal variability, in particular, the AMO, as seen in the consistent interdecadal variability in AMO in Fig.R1.1. We have also added AMO series in Fig.1d (orange). Although the MMEM does not show a strong weakening between 1950s and 1980s (partly due to the filtering of internal variability by using MMEM), it is likely that such weakening did occur in the real world. In this case, the post-1980s weakening in our SSS index is likely indicative of a weakening AMOC since the 1950s. However, as we mentioned around **Line 68-70**, in T_NA, this multi-decadal weakening is contributed more by internal variability (e.g. AMO) with the anthropogenic signal largely overwhelmed. In our salinity fingerprint, this weakening is likely contributed more by the anthropogenic weakening (please see our argument on the trend/variability ratio). If we are correct, in the future, the post-1980 strengthening signal

that is so strong in T_{NA} will be weaker in our salinity fingerprint. This can only be verified in future observations.

Fig.R1.1 (a) Comparison between SST based AMOC index and AMO index. Thin and thick lines represent annual and 11-yr running mean series. (b) as (a) but for detrended AMOC index. AMO index is defined as the weighted mean SST over the North Atlantic (0° N to 60° N) with the global mean SST (60° S to 60° N) removed.

Furthermore, as we discussed in the text, a decrease in T_{NA} index does not necessarily indicate a weakening in AMOC, as the warming hole between 1950 and 1990 can be also driven by atmospheric processes without an explicit role of AMOC change (Line 74-77 and Line 244-246). Therefore, there can be no AMOC-related SSS change in response to the post-50s cooling in the SPNA.

In summary, we think the relationship between T_{NA} index and AMOC is not so reliable, and T_{NA} index has small signal-to-noise ratio, which make it difficult for detecting long-term forced AMOC change. In comparison, the declining trend in salinity fingerprint is more likely

an indication of AMOC weakening, starting sometime earlier than the 1980s.

4. How is the SSS index better than a low frequency filtered T_{NA} index (which would also reduce the noise?) Just because the signal-to-noise ratio in the SSS index is better, this does not mean that it is a better representation of the actual AMOC (as the AMOC might display large internal variability).

R1.4: A low-pass filter of T_{NA} may also filter out internal variability. However, this filtering approach on T_{NA} has two disadvantages. First, given the long-time scale of the low-pass filter needed to filter out interdecadal variability (say, 50 years, comparable for the AMO), the filtering creates a big end point problem on the short available record, which is almost of comparable length as the filter. This end point problem is especially problematic because the signal (global warming forced change) is near the end point. Second, as discussed earlier, T_{NA} is also subject to atmospheric forcing that is unrelated to AMOC changes. In contrast, our salinity fingerprint does the filtering naturally through dynamics (therefore there is no end point problem), and is less directly affected by the atmospheric forcing.

5. Why is there this discrepancy between observed SSS indices and modelled SSS indices (figure 1b,c). If the SSS index mainly picks up the forced AMOC signal, shouldn't it be very similar to the MEMEM SSS indices?

R1.5: We think it is mainly due to the different climate sensitivity of SSS in response to AMOC change (wide ranges of responses in models). As a single realization, the observation (solid black line) falls well into the model spread (scatter) as seen in Fig.2b and d.

General remarks:

When you talk about AMOC strength in the models: why do you use this more flexible definition instead of looking at 26N or 30N AMOC to make it better comparable to the observations? The same question goes for the South Atlantic AMOC, looking at 34.5S to make it comparable to SAMBA might be nice.

R1.6: There are several reasons. First, our choice mainly follows the traditional definition (maximum overturning streamfunction below 300-m over 30°-50°N in the Atlantic). Second, our conclusion is not sensitive to the choice. Third, the direct observations are too short for

model-data comparison, especially, the 26.5°N RAPID AMOC measurement. The robustness of our conclusion regardless of the definition of AMOC intensity can be seen below. For models (CMIP5&6), the AMOC transport with flexible definition is generally larger than that at 26°N (Fig.R1.2a,b). But the variability of AMOC is very similar between the two definitions (Fig.R1.2c,d). The CMIP6 model mean 26°N AMOC is comparable to RAPID (~17Sv, dark blue) averaged over the overlapping period. RAPID AMOC shows strong decadal variability, which is absent in MEMM AMOC. The post-80s change in model mean AMOC (2-3Sv) agrees well with the ECCO reanalysis. For the South Atlantic, the model mean SAMOC intensity at 35°S is about 18Sv, also comparable to the reconstructed SAMOC using Argo and altimetry averaged over the overlapping period (Fig.R1.3a,b). Distorted by strong interannual variability, the trend of reconstructed SAMOC is very weak and insignificant. The weakening trend of modeled SAMOC is more consistent with the ECCO reanalysis, with latter of a larger magnitude (Fig.R1.3c,d).

Fig.R1.2 Model-data comparison for *northern* AMOC intensity. (a) 26°N AMOC; (b) AMOC maximum in the North Atlantic.(c-d) As (a-b) but for anomaly.

Fig.R1.3 Model-data comparison for *southern* AMOC intensity. (a) 34°S AMOC; (b) AMOC maximum in the South Atlantic.(c-d) As (a-b) but for anomaly.

For your definition of T_{NA} you use annual data (instead of Nov-May SST) and a larger region than ref.34 – this of course could be one of the reasons why you increase the noise in the index, as your paper concentrates on finding the best signal-to-noise ratio, this is rather unfortunate.

While I understand that one could argue for a larger region, especially when looking at models as the region most sensitive to AMOC heat transport changes might differ from model to model, I strongly suggest also only using Nov-May or at least the cold season SST to minimize the more chaotic influence of the atmosphere on the index.

R1.7: Thanks for the suggestions. The region used to calculate the T_{NA} is actually [15°W-40°W,46°N-60°N] and we apologize for the mistake ([20°W-55°W,46°N-60°N]) in the original submission. Following the suggestion of the reviewer, below we compared T_{NA} indices using Nov-May mean and annual mean SST. Fig.R1.4 shows the comparison for observational data. It is true that the difference between SPNA SST and global mean SST is larger in cold season compared to the annual mean (by ~1.5K; Fig.R1.4a), with the signal(trend)-to-noise(variability) ratio (T/V ratio) being only 0.1 higher for cold season (Fig.R1.4b). For models, there is also no significant improvement in the signal-to-noise ratio (Fig.R1.5). Further, we have tested the trend correlation between cold-season SST/annual SST and AMOC during 1850-1985 and found the

correlation is only slightly increased while still largely reduced across model ensemble means compared with that across model members (Fig.R1.6). Therefore, we added a comment on the robustness of using cold season T_{NA} around **Line 140-141**. “This result remains robust if the annual T_{NA} is replaced with the cold season T_{NA} ”.

Fig.R1.4 Time series (a) and T/V ratio (b) of SST based AMOC indices using annual mean (green) and cold season (blue) SST. In (b) T/V ratio as a function of starting year from which the T/V is calculated (all ending in 2021).

Fig.R1.5 As Fig.S6 but with SST index using cold season SST is included (green diamond).

Fig.R1.6 (a-b) same as Fig.2(e-f). (c-d) as (a-b) but for cold season SST index.

Further comments:

ll. 64ff/Figure 1d) This shows the CMIP5 and CMIP6 T_NA is very different from the observed one, as the former do not show a long-term weakening over the 20th century. I assume MMEM is multi-model ensemble mean? This should be defined.

R1.8: Defined.

ll.69-70 “strong strengthening” sounds strange, maybe rephrase

R1.9: We rephrased to “strengthening”

ll.70-71 Figure 1d does not show the AMV, if you refer to the AMV here it would be good to show, also what do you mean when you say the T_NA is overwhelmed by the AMV, that it drives most of the observed signal? As AMV likely contains an AMOC signal this an influence of AMV on T_NA does not exclude the AMOC as the main T_NA driver.

R1.10: Good point. We have added the AMO index (which is similar to but more used than AMV) in Fig.1d and specify its definition in the Methods (also see Fig.R1.1). Here, T_NA is actually a mixture of AMO and a weakening trend (Caesar et al., 2018; Fig.R1.1). It is true that AMO likely contains an AMOC signal, but what we focused on here is the long-term forced AMOC trend and AMO is therefore regarded as “noise”. We can see that the trend signal of T_NA is largely overwhelmed by the AMO “noise” (i.e. the trend signal is small compared to the amplitude of AMO) (Fig.R1.1).

ll. 90 ff “implying a forced AMOC response likely of little to slight weakening trend before 1980s” – this is speculative maybe rephrase this?

R1.11: We have deleted this sentence and rephrased as “*The difference between the two MMEMs is likely caused by aerosol forcing, which remains highly uncertain²⁸ and most likely overestimated in CMIP6^{27,29} (Methods)*” (Line 88-90).

ll. 93 ff Can you be more specific about the trends in CMIP5/6 after and before 1980 (i.e. is this the linear trend over 40 years? If so, it would be great to compare it to the 1920-1980 trends).

R1.12: The global mean sea surface warming rate during 1920-1980 is $0.001^{\circ}\text{C}/\text{yr}$ and $0.002^{\circ}\text{C}/\text{yr}$ for CMIP6 MMEM and CMIP5 MMEM, respectively. The smaller rate for CMIP6 may be caused by the overestimation of aerosol cooling effect. The corresponding AMOC weakening rate is $0.014\text{ Sv}/\text{yr}$ ($0.7\text{Sv}/50\text{years}$) and $-0.002\text{ Sv}/\text{yr}$ ($-0.1\text{Sv}/50\text{years}$) for CMIP6 MMEM and CMIP5 MMEM, respectively. In comparison, the warming rate after 1980 is stronger, with $\sim 0.016^{\circ}\text{C}/\text{yr}$ and $\sim 0.015^{\circ}\text{C}/\text{yr}$ (both $\sim 0.8^{\circ}\text{C}/50\text{yrs}$) for CMIP6 MMEM and CMIP5 MMEM, respectively. Corresponding to the stronger warming, there is an accelerated AMOC weakening after 1980, with $-0.08\text{ Sv}/\text{yr}$ ($-4.0\text{Sv}/50\text{years}$) and $-0.023\text{ Sv}/\text{yr}$ ($-1.2\text{Sv}/50\text{years}$) for CMIP6 MMEM and CMIP5 MMEM, respectively. We have also mentioned these values in the text around Line 87-95.

ll. 96 ff/ Figure 1a I am a little surprised about the GloSea5 data (purple line), as the new Jackson et al., 2022 paper shows that basically all AMOC reconstructions hint at a strengthening of the AMOC from the 80s to late 90s (including the GloSea5 data) – can you explain this discrepancy?

R1.13: The purple line in Figure.1a represents the latest ocean state estimate ECCOv4r4 (version 4 release 4), which reproduce observations in a physically and statistically consistent manner (Forget et al., 2015). We found the weakening in MMEM AMOC matches especially well with the linear change of AMOC during the overlapping period 1992-2017. We also plotted the ECCOv4r4 AMOC across different latitudes and found the weakening trend seems to be a basin-scale signal (Fig.R1.7). GlobSea5 data is shown in Figure.S2 with grey line. In their Fig.2 of Jackson et al.(2022), none of the AMOC reconstructions started earlier than 1990 and generally show a decreasing trend since 1990, especially for subpolar AMOC reconstructions (in their Fig.2a-e). It is likely that the meridional coherence of AMOC can be degraded by wind forcing, leading to opposite AMOC change between subtropical and subpolar regions on interannual to decadal timescale (Fig.R1.7a green vs. blue lines; Gu et al., 2020). In contrast, buoyancy-forced subpolar AMOC signal, such as that triggered by anthropogenic warming, can spread southward coherently (refs.35-37). This anthropogenic long-term AMOC change is our focus indeed.

Fig.R1.7 ECCOv4r4 AMOCs at different latitudes.

References:

Forget, G., et al. ECCO version 4: An integrated framework for non-linear inverse modeling and global ocean state estimation. *Geoscientific Model Development* **8**, 3071-3104 (2015). <https://www.geosci-model-dev.net/8/3071/2015/>.

Jackson, L. C., et al. The evolution of the North Atlantic Meridional Overturning Circulation since 1980. *Nat. Rev. Earth. Environ.* (2022). <https://doi.org/10.1038/s43017-022-00263-2>.

Gu, S., Liu, Z., Wu, L. Timescale dependence of the meridional coherence of Atlantic Meridional Overturning Circulation. *J. Geophys. Res. Oceans* **125**, e2019JC015838 (2020). <https://doi.org/10.1029/2019JC015838>.

Il.114ff If the indices behave similarly, does this means S_STSIP is approximately constant or just very small? Please comment. Plus, in figure 2 the results/correlation coefficients are actually quite different for the two indices, not supporting that they behave very similar.

R1.14: Yes, the change in S_STSIP is small compared with S_STSA. As we mentioned in the text and in Zhu and Liu (2020), the S_STSA index (S_SA) contains both AMOC signal and surface freshwater signal (precipitation minus evaporation, E-P). For the historical period 1850-1985 as shown in Figure 2, the correlation between S_SA trend and AMOC trend is higher, because E-P influences S_SA in the same direction as AMOC does. That is, global warming causes AMOC weakening and subtropical salinification, both leading to an increase in S_SA. In contrast, by subtracting S_STSIP from S_SA, the S_S index contains some noise from the Indo-Pacific variability while, to some extent, reduces the quasi-zonally uniform (E-P) influence. We therefore argue that S_SA generally has higher correlation with AMOC compared with S_S while S_S is more directly related to AMOC change. For lead-lag correlation in control and forced experiments shown in Fig.3 and Fig.S3, these two indices indeed behavior similarly.

References:

Zhu, C., Liu, Z. Weakening Atlantic overturning circulation causes South Atlantic salinity pile-up. *Nature Climate Change* **10**: 998-1003(2020).

ll.117ff I am not sure whether I agree: Why do you assume that the remote response can only be seen in your fingerprint?

R1.15: All the previous AMOC fingerprints are located locally in the North Atlantic. Our fingerprint is unique in that it locates in the South Atlantic and represents the remote AMOC response. We rephrased this sentence as “In contrast to all previous AMOC fingerprints *that are located locally in the North Atlantic...*” (Line 116-117).

ll.120 ff Could you give the correlation between AMOC and SSS index for the different models (both cross-member and cross-model) to support this?

R1.16: The trend correlations have been shown in Figure.2. In each panel, red and blue numbers represent the correlation coefficients for CMIP5 and CMIP6, respectively.

ll. 135 ff While it is true that there seems to be more forced “signal” in the SSS fingerprints compared to the T_NA the actual correlation between AMOC and index is for individual

members higher for T_NA. This indicates that T_NA is a better representator of the actual AMOC (which of course can contain internal variability).

R1.17: We partially agree. As we replied in R1.4, the T_NA is likely a better representator of the actual AMOC for variability, but not for the long-term anthropogenic trend . The good representation of T_NA on variability is seen clearly in the high correlation between T_NA and AMOC in control simulations. However, it is exactly this good representation of internal variability that is, we think, unfortunately a disadvantage for detecting the long-term anthropogenic trend. This follows, as discussed earlier, the internal variability is the “noise” that can overwhelm the long-term trend “signal”. This is one reason that our the SSS index is optimal for detecting the trend, because it “filters out” variability dynamically and therefore has higher (trend) signal-to-(variability) noise ratio. The other reason our SSS index being optimal is that it is less sensitive to radiative forcing than T_NA.

ll.150 ff Why does a higher correlation and a smaller lead time correspond to more noise/internal variability?

R1.18: This is in the control run that has no anthropogenic forced trend. The higher correlation between T_NA and AMOC in PI control simulations suggests that T_NA is better in recording AMOC internal variability (which is the “noise” to anthropogenic forced signal) than the salinity fingerprint.

ll. 164 ff Why does the lag between AMOC and SSS index change for the different simulations (from about 20 years in piControl to only a ~1-2 years in Hist-GHG)? What kind of mechanism could cause this, and don't you think that this big change could be indicating that the SSS index does not work that well??

R1.19: Good question. The reason for the short time lag in Hist-GHG is that both AMOC and SSS index in Hist-GHG run contain a strong global warming trend (Figure.4a,b), which will shorten the lag. When the long-term trend is removed, the time lag becomes 10- 20 years (Fig.R1.8), similar to that in piControl simulations (Fig.3a). Interestingly, this issue does not occur to Hist-aer. Unlike Hist-GHG, AMOC in Hist-aer run contains strong multi-decadal variability around 1980s (Fig.4a) and therefore there is a clear time lag in Hist-aer.

Fig.R1.8 As Fig.3(b) but for detrended time series.

II. 174 ff While you convincingly show that changes E-P hardly affect the SSS indices, this does not mean that the changes have to be due to AMOC changes. They could also relate to changes in the gyre circulation or other local ocean circulation changes – how do you know that that is not the case?

R1.20: Great question. We thank the reviewer for the insightful comments. Indeed, the wind effect and the related gyre circulation change is difficult to isolate in a fully coupled model simulation, if not impossible. Nevertheless, we think the trend signal is forced mainly by the buoyancy forcing, instead of wind. First, our selection of the AMOC intensity as below 300-m helps to avoid the response of the wind-driven subtropical cell. Second, in an attempt to isolate the wind effect, we checked the change of zonal wind stress under CMIP5 RCP45 (relative to historical run) as shown in Figure R1.9. In the South Atlantic, there are positive (negative) zonal wind stress anomaly around the northern (southern) boundary of STSA domain. This anomalous wind stress pattern will cause anomalous Ekman transport exporting saltier water from subtropics and leads to surface freshening within S_S domain, opposing to the surface salinity pileup. This surface Ekman transport alone may therefore contribute negatively to surface salinity pileup. Third, the wind stress is intensified over the Southern Ocean. In the steady state

theory, this should strengthen, as opposed to reduce, the AMOC (albeit after a long adjustment time scale (e.g. Allison et al., 2011)). Indeed, we speculate the wind effect may be one reason that causes the spread of correlation between S_S and AMOC in models.

Fig.R1.9 Atlantic zonal wind stress changes (unit: Pa) under anthropogenic forcing in CMIP5 models. (a) Zonal wind stress climatology in historical run (averaged over year 1980-2005). (b) Same as (a) but for climatology in RCP4.5 scenario (averaged over year 2080-2100). (c) Difference between (b) and (a). Black thick line in each panel is the ensemble mean result.

In our 2020 paper (Zhu and Liu, 2020), this difficulty to isolate the wind effect in a coupled model is indeed one major motivation for us to carry out the ocean-alone sensitivity experiments, in all of which the wind stress remains unchanged and therefore ocean circulation change is contributed purely by the AMOC. In those experiments, when the AMOC weakens, there is always salinity pile-up in the South Atlantic. The salinity pile-up has also been widely reproduced in water-hosing experiments (e.g., Stouffer et al., 2006). Therefore, we think the salinity pile-up is mainly related to the AMOC weakening.

References:

- Allison, L. C., Johnson, H. L., Marshall, D. P. Spin-up and adjustment of the Antarctic Circumpolar Current and global pycnocline. *J. Mar. Res.*, **69**, 167–189 (2011).
- Stouffer, R. J., et al. 2006: Investigating the causes of the response of the thermohaline circulation to past and future climate changes. *J. Clim.*, **19**(8):1365–1387 (2006).
- Zhu, C., Liu, Z. Weakening Atlantic overturning circulation causes South Atlantic salinity pile-up. *Nature Climate Change* **10**: 998-1003(2020).

II. 181 ff It seems to me that the SSS indices cannot capture this final response to GHG and

aerosols consisting of an increase in AMOC strength in the 80s and following decrease, either – even though the signal has a fairly high amplitude. Did you expect this?

R1.21: We agree that the trend reversal signal in AMOC is not captured well by the SSS indices (S_{SA} is better than S_s in capturing this signal). We think it is due partly to the time lag between the SSS indices and the AMOC. As seen more clearly in the Hist-aer, in contrast to an increase in AMOC before the 1980s and a following decrease, the SSS indices show only increase during the whole period, although with weaker magnitude after the 1980s. This increase in SSS indices opposes the decrease in Hist-GHG, leading to the small change in the historical run.

ll. 191 ff You are citing a study about a 2-layer model, can you provide evidence that the real world AMOC response is set by Rossby waves and give an estimate of the expected time lags?

R1.22: This is an interesting question. Most OGCM studies on the meridional coherence of AMOC seem to focus on variability, mainly using the lagged response, for example in ref. 38 (Gu et al, 2020) and the references therein. These lagged correlation analyses suggest a time lag of 5-10 years. However, the salinity fingerprint may further lag the AMOC in the South Atlantic, because it takes time for the AMOC to affect basin-mean salinity, which leads to a lagged time of about 20 years, as shown in our correlations analysis in Fig.3 for control and other runs (also see R1.19 above). We are not aware of publications that explicitly studied the lag and wave adjustment process for the real-world case in response to anthropogenic forcing, especially related to the salinity signal. This is something we would like to explore in the future.

References:

Gu, S., Liu, Z., Wu, L. Timescale dependence of the meridional coherence of Atlantic Meridional Overturning Circulation. *J. Geophys. Res. Oceans* **125**, e2019JC015838 (2020). <https://doi.org/10.1029/2019JC015838>.

ll. 244 ff Why is this accelerated slowdown that is, as you state, predicted by CMIP5/6 not visible in the models' SSS indices (Figure 1b,c)? That does not make sense.

R1.23: As we replied in R1.21, we think it is due to the time lag between the SSS indices and the AMOC. Also, model responses associated with the time lag feature have a large spread and therefore the MEM over a specific period does not capture the accelerated slowdown well.

Figure 5/S6 Can you comment on why the T/V ratio in all models seem to show a minimum right at the location/latitude of the T_{NA} index? This made me curious.

R1.24: We speculate that there is high-frequency atmospheric variability (may related to storms) that contributes to both low T/V ratio in T_{NA} and subpolar AMOC in GISS model. We also note that the T/V ratio profile can be model dependent.

References

Worthington, E. L., Moat, B. I., Smeed, D. A., Mecking, J. V., Marsh, R., and McCarthy, G. D.: A 30-year reconstruction of the Atlantic meridional overturning circulation shows no decline, *Ocean Sci.*, 17, 285–299, <https://doi.org/10.5194/os-17-285-2021>, 2021.

Fu et al., 2020: <https://doi.org/10.1126/sciadv.abc7836>

Jackson et al., 2022: <https://www.nature.com/articles/s43017-022-00263-2>

Reviewer #2 (Remarks to the Author):

Review of “Accelerated Slowdown of Atlantic Circulation Emerged in Optimal Fingerprint” by Chenyu Zhu and colleagues.

This manuscript addresses the question, “Has AMOC slowed down over the last century”? The topic is timely, controversial and of considerable scientific interest. The authors are building off of the work by two of the present authors (<https://doi.org/10.1038/s41558-020-0897-7>) where they identify a “fingerprint” of AMOC variability in the salinity of the South Atlantic, specifically the difference in salinity between the tropical to subtropical South Atlantic (10S-34S) and the Indo-Pacific (10S-34S). In the current work, they apply that fingerprint using EN4 salinity data and argue, based on several lines of evidence that the southern hemisphere salinity proxy is better than a North Atlantic “warming hole” index for monitoring decadal to centennial-scale trends in AMOC.

I think the salinity index idea is exciting, novel, and has promise for getting us closer to understanding the temporal development of AMOC over the last century. The idea is a substantial leap forward for which the authors should be congratulated and deserves to be published. I am recommending major revisions because of three factors. First, I have deep reservations about the use of the EN4 salinity data for the analysis of salinity trends, given that salinity observational datasets tend to be sparse in both the South Atlantic and tropical South Pacific. Since these data form the primary results referred to in the title, it is important that they be robust. However, I also note that a great deal of the work described by this manuscript involves using model data to better understand why the salinity data gives a better perspective on the multidecadal trend of AMOC than does a North Atlantic sub-polar gyre-based index. If the EN4 data turn out to be unusable in the early period of the analysis, I think it would be very reasonable to change the focus of the paper to an analysis of why the salinity proxy of AMOC captures multidecadal trends in AMOC better than North Atlantic temperature proxies of AMOC.

The second aspect of the current manuscript that gives me pause is that the text never really

explains how the salinity build-up is caused by AMOC slow-down. I understand the current work is building on the prior paper, but even reading the prior paper, I was disappointed with the lack of a mechanistic explanation in either paper. What causes the salinity convergence or divergence in the southern hemisphere surface? I would expect some explanation involving salty Agulhas Leakage, northward low salinity water crossing the equator, and air-sea freshwater fluxes, but the current manuscript lacks even the briefest of explanations along similar mechanistic lines. Since this is the basis for the paper, I think it is worth addressing.

The third factor that must be addressed before publication is the writing itself. The writing is often very unclear because of the lack of precision in the use of words and sometimes the incorrect usage of words. Often statements are ambiguous as written. I could guess from the context what the authors intended, but this needs to be corrected before publication.

Additionally, the manuscript is littered with grammar errors, which can be relatively easily corrected. Both of these things combine to make the manuscript very difficult to read and comprehend. I give only examples in the specific comments because it became rather tedious to point out every grammar mistake or ambiguous statement and my job is to review the science, not copy edit the manuscript.

Response to the overall comment: We appreciate the Reviewer for providing these constructive comments, particularly on the data uncertainty and writing. We have fully improved the manuscript. We copied the original review in blue and write our reply in black. Changes in the manuscript text are shown with red color highlighting.

Some specific comments for improving the manuscript are listed below.

SPECIFIC COMMENTS

More on Trends in EN4 data and data coverage.

It is unclear to me if the authors are using the “raw” data or the infilled and gridded “analysis” data of EN4. I assume the gridded analysis field. EN4 relaxes to a 1970-2000 climatology when data are sparse. Given that salinity observations tend to be sparse, I think it is vital for any interpretation of long-term trends to check the observation weights in the data. It is worth a line on the main figure (figure 1) that demonstrates that the data are based on some measurements

and not just relaxed to climatology (for example, plotting the number of observations in the region, or % of grid boxes where the weight is majority on the data rather than climatology).

I say this because the EN4 Product Users Guide specifically cautions against analysis of trends in data sparse areas. Relevant text is reproduced from the guide available at the link

below: https://www.metoffice.gov.uk/hadobs/en4/EN.4.2.2_Product_User_Guide_v1.0.pdf

“6.5 CAN I USE EN4 FOR TREND ANALYSIS?”

EN4 is an observation based [sic] data set and therefore coverage varies in both time and space. For the complete fields provided in the analyses we strongly encourage users to look at the ‘observation weights’ variables. These will inform users how much the analysis value has been influenced by observations (values closer to one) and how much it has been determined by background fields (values closer to zero). We would not encourage the use of EN4 analyses for trend analysis in areas where the observation weights are low.”

R2.1: We appreciate the Reviewer for pointing the data issue. The availability of salinity observations in the South Atlantic is indeed an issue for our fingerprint, and certainly a disadvantage relative to T_{NA}. Nevertheless, the lack of variability in the earlier part is not just in our salinity fingerprint, but also in T_{NA}. Therefore, it is possible that the stronger trend/variability in the later half relative to the earlier half is still valid. In other words, if in the early part, had the T_{NA} showed a significant change (trend or variability) while our salinity fingerprint does not, there will be a serious problem in our interpretation of the accelerated weakening in terms of our salinity fingerprint.

As for the data set, in the present study, we are using analysis data of EN4 (1900-2017). In the revision, salinity from 1900 to 2021 from the latest version EN4.2.2 is used (EN4.2.1 in our first submission). Using EN4.2.2 output, we plotted the number of observations and the mean weight over the South Atlantic (Fig.R2.1). As the reviewer mentioned, there are rare salinity observations before 1950s and the data becomes plentiful only after 2000s (Argo era). In the revised version, we have weakened our conclusion by adding “likely” in front of acceleration in the title and many other places in the text. In an attempt to evaluate the significance of the trend, we have also checked another salinity data from Institute of Atmospheric Physics (IAP), China. Compared with previous salinity products, this product “overcome some of the inconsistencies

present in existing salinity reconstructions by using an interpolation technique that uses information on the spatiotemporal covariability of salinity taken from model simulations” and is shown to be “more reliable for examining long-term salinity changes” (Cheng et al., 2020). The SSS index using IAP salinity data is shown in Fig.R2.2 along with 90% CI (sampling error provided by Dr. Lijing Cheng from IAP). The decreasing trend in IAP SSS index is shown to be significant since the late 1970s and the accelerated decreasing in 1990s seems to be also consistent with that in EN4 and Ishii data. Therefore, although with notable data uncertainty in the earlier period, we think the stronger trend in the later period may be still valid. We have added some discussions on this issue around Line 129-132.

Fig.R2.1 Number of salinity observations (upper) and mean observation weight (lower) in the South Atlantic on a monthly basis from 1900 to 2021.

Fig.R2.2 IAP salinity index.(a) S_s index; (b) STSA mean SSS; (c) STSIP mean SSS. Shading denotes 90% CI.

References:

Cheng, L., et al.2020: Improved Estimates of Changes in Upper Ocean Salinity and the Hydrological Cycle. *J.Clim.*, **33**:10357-10381.

Line 43: “the collaborative forcing” should probably read “combined forcing” since “collaborative” indicates people working together and is not used for inanimate forces.

R2.2: Corrected.

Line 49 “continually collaborative anthropogenic forcing” see above for proper use of “collaborative”.

R2.3: Corrected.

Line 59-60. The text and references imply that the high latitude North Atlantic is the only target for AMOC reconstructions, but Rong Zhang developed a “fingerprint” for AMOC in the equatorial Atlantic. I think just adding in this reference would be enough to address the issue, because technically, the low-latitude fingerprint is also in the North Atlantic.

Zhang R (2007) Anticorrelated multidecadal variations between surface and subsurface tropical North Atlantic. *Geophys Res Lett* 34:1–6.

R2.4: Thanks for pointing that out. We added the reference of Zhang (2007) in the revised manuscript.

Line 68 “An application of TNA to real world has been” is one of many examples where the word “the” is missing. It should read “An application of TNA to the real world has been”

R2.5: Corrected.

Line 86 “Before 1980s” should be “Before the 1980s”. This is consistently in error throughout the paper.

R2.6: Corrected. We have modified the same error throughout the paper.

Line 89 “is highly uncertainty” should read “is highly uncertain”

R2.7: Corrected.

Section starting on Line 100, “Optimal Fingerprint...” This section is where I think the reader needs a more mechanistic explanation. Since this is the foundation of the current work, I think it is perhaps worth a figure, but certainly a few sentences to explain it better. I have to admit the explanation in the author’s previous paper sounded to me more like an assertion rather than a reasoned explanation and the “mechanism schematic” buried in the supplementary text was overly simplistic to the point of not being very helpful. I caution the authors to not repeat earlier mistakes.

R2.8: Thanks for pointing this issue. Now in the revised version, we have added a mechanistic explanation in **Line 119-123** as “*In response to a weakening AMOC, the northward salinity transport in the upper South Atlantic is reduced. Given the increase of mean climatological salinity from the subpolar to subtropics, the reduction in salinity transport is greater downstream (northern side of STSA domain) than upstream (southern side of STSA domain) in the upper branch of the AMOC, leading to a salinity pile-up in the South Atlantic³⁵*”.

Line 115: As a reader, I'm not convinced at this point in the text that $S_s = S_{sa}$ as asserted here. Because of this, the associated parenthetical phrase sounds more like sloppiness than rigorous science. I think it is worth being specific about which (S_s or S_{sa}) you are using throughout the paper because although by the end of the paper, I would agree with you, the reader has no reason to believe the authors at this point in the text.

R2.9: Thanks for the suggestion. We have deleted the parenthetical phrase and specified which one is used throughout the paper in the revised version.

Line 119-120 “SPNA deep-convection region, which is known to force AMOC signal propagating southward coherently” While I don’t argue that deep convection creates a “push” for AMOC, I see no reason that the trend or variability couldn’t be also caused by the pull of upwelling/diapycnal mixing (see Visbeck, M. (2007), Power of pull, Nature, 447(7143), 383, doi:10.1038/447383a. and two recent papers by Robbie Toggweiler <https://doi.org/10.1029/2018JC014794> and <https://doi.org/10.1029/2018JC014795>.

R2.10: Thanks for pointing this issue. We should have made it clearer. Here “which” refers to *buoyancy flux forcing* in the SPNA deep-convection region. Model simulations show that the AMOC slowdown under global warming is mainly forced by a slowly increasing surface heat flux/freshwater flux over the subpolar North Atlantic (Gregory et al., 2005; Visbeck 2007). This type of buoyancy-forced AMOC response has been shown to propagate southward coherently against the distortion by variable wind forcing (ref.36-38). We have rephrased this sentence as “*In contrast to all previous AMOC fingerprints that are located locally in the North Atlantic, our SSS index is unique in that it represents a remote AMOC response to the buoyancy flux forcing in the SPNA deep-convection region, which is known to force AMOC signal propagating southward coherently*” (Line 116-119).

The wind in the Southern Ocean (SO) is projected to intensify in response to global warming. This should lead to an increase, instead of decrease, of the AMOC; furthermore, this adjustment process may take a longer time, likely over a hundred years (Visbeck 2007; Allison et al., 2011). See our reply to Reviewer 1 (R1.21, Fig.R1.9) for relevant issues. Therefore, we don’t feel the SO wind is active yet in this transient response process. This does provide an interesting topic for our future studies.

Reference:

Allison, L. C., Johnson, H. L., Marshall, D. P. Spin-up and adjustment of the Antarctic Circumpolar Current and global pycnocline. *J. Mar. Res.*, **69**, 167–189 (2011).

Gregory, J. M. et al. 2005: A model intercomparison of changes in the Atlantic thermohaline circulation in response to increasing atmospheric CO₂ concentration. *Geophys. Res. Lett.* **32**, 112703.

Line 132: Sentence “This hypothesis is supported by multiple evidences and, more importantly, is underpinned by fundamental dynamics, as discussed below.” As the reader reads the first part of this sentence, they may expect this to be a summary of the above section which has not brought forth the multiple evidences. It is not until the end of the sentence that the reader learns this sentence is transitioning to the next section. Please change to something like “ The following section describes the multiple lines of evidence and the fundamental dynamics underpinning this hypothesis.

R2.11: Thanks for the suggestion. We have changed to “*The following section describes the multiple lines of evidence and the fundamental dynamics underpinning this hypothesis*”.

Line 136-137 this sentence is referring to model data and should describe it as such.

R2.12: We have changed the sentence to “Our first *modelling* evidence...”.

Lines 164-166 This sentence is a great example of the lack of clear communication in the writing that makes this manuscript frustrating to read. “The lead-lag correlation with AMOC is more than doubled for the SSS fingerprint (Fig.3 b, c vs a), such that the correlation for SSS index becomes comparable or even greater than that for TNA.” The ideas I think the authors are trying to communicate are more effectively stated in this sentence,” The maximum lagged correlation between AMOC and SSS-based AMOC indices nearly doubles in forced model runs compared with the pre-industrial control run and becomes comparable to that for TNA.” Note that the two data sets being correlated (AMOC and SSS fingerprint) are specified and the source of the data, which is relevant for understanding the result, is also specified. Lack of such specificity in the writing generally makes this paper more difficult to read than necessary.

R2.13: Thanks for the comment. We have improved the readability of our paper. For this sentence, we have changed to “*The maximum lagged correlation between AMOC and SSS-based AMOC indices nearly doubles in forced model runs compared with the pre-industrial control run and becomes comparable to that for T_{NA}* ” (Line 172-174).

Lines 174-176. “The SSS response to GHGs or aerosols forcing, especially Ss, is forced completely by the AMOC change, instead of surface E-P forcing associated with the hydrology

response³⁴”. This is a very strong assertion with little evidence and no stated mechanism. I understand that you are citing the prior work, but all I got out of reading reference 34 were some vague comparisons between the patterns of E-P trends vs the patterns of SSS trends and some more assertions that it is AMOC-related. This is a key concept in your paper and should not be treated so lightly.

R2.14: In our previous paper (Zhu and Liu, 2020), the direct comparison between patterns of E-P trends vs the patterns of SSS trends just served as a motivation, which made us speculate that AMOC may have a role on the salinity pile-up. We have then made many efforts examining the relation between AMOC/E-P and SSS in climate models in both statistical analysis and sensitivity experiments. For coupled simulations, we showed: 1) interannual correlation between AMOC and SSS index is higher than that between E-P and SSS index; 2) trend correlation between AMOC and SSS index across different warming scenarios (historical, rcp45 and rcp85) is even higher while the correlation between SSS and E-P becomes even negative. In ocean-alone sensitivity experiments and related salinity budget analysis, we further showed that pure E-P forcing (derived from RCP45 anomaly) forces nearly equal salinification between the South Indo-Pacific and the South Atlantic; only with a weakening AMOC, can the salinity pile-up take place in the South Atlantic relative to the Indo-Pacific. We have also discussed this issue in Supplementary Text along with Fig.S4 and S5. Please also see our reply to Reviewer 3 for more details (R3.12).

Line 177. “even after MMEM” grammar issue. Should read something like, “even using the MMEM”

R2.15: Corrected.

Line 178-179 “When all the forcing combined, the competition of GHGs and aerosols impacts prior to 1980s lead to a prevalence of natural AMOC variability”. The sentence has multiple grammar issues. It should read something like, “When all the forcing is combined, the competition between the impacts of GHGs and aerosols prior to the 1980s leads to the prevalence of natural AMOC variability.”

R2.16: Corrected.

Line 190 “trend-signal/variability-noise ratio” At this point in the manuscript the reader is familiar with the fact that the authors consider the trend to be the signal and the variability to be the noise. This can be simplified here and below as trend/variability ratio so that the text remains specific.

R2.17: Thanks. We have simplified the term.

Line 191 “to climate forcing.” Please be specific about what climate forcing. I think the authors mean to indicate radiative forcing, but there are other climatic forcing components of AMOC (e.g., winds and freshwater fluxes). Again, this is an example of the imprecise use of language.

R2.18: We have changed to “to buoyancy forcing over the SPNA”.

Line 191-193. This blithely assumes the "push" is more important than upwelling/diapycnal mixing "pull" in AMOC adjustment. I'm not sure this is correct. I think it is worth a sentence justifying this perspective. See Visbeck, M. Power of pull. Nature 447, 383 (2007). <https://doi.org/10.1038/447383a> and two papers by Toggweiler et al 2019 10.1029/2018JC014794 and 0.1029/2018JC014795.

R2.19: In the present study, we focus on the AMOC change (instead of the mean state) under current global warming, which has been shown to be influenced mainly by the buoyancy forcing, such as heat flux forcing and freshwater flux forcing, over the SPNA. Please see R2.10 for more information.

Line 250 I caution against the use of "emerge" because it indicates “climate emergence” which has a specific meaning not intended here.

R2.20: We have replaced “emerged” with “occurred”.

Line 445 “Our estimate of aerosol forcing is in good agreement with the gridded aerosol community datasets (CEDs) with $\sim 0.3\text{W/m}^2$ stronger, which...” The “with $\sim 0.3\text{W/m}^2$ stronger” is awkward at best and grammatically wrong at worst. The sentence is long and might be better if split into two sentences.

R2.21: Thanks for the suggestion. We have split the sentence into two: “*Our estimate of aerosol*

forcing agrees well with the gridded aerosol community datasets (CEDS)^{30,61}. The larger magnitude (by $\sim 0.3 \text{ W/m}^2$) in our estimate compared with CEDS may be attributed partly to the overestimation of aerosol forcing in CMIP6 models and partly to the climate feedbacks in models, notably the sea ice-albedo feedback (i.e., more reflection of incoming short-wave radiation by increasing sea ice in response to aerosol's cooling effect)."

Line 557 I am concerned by defining AMOC without the surface layers when much of the return flow to the North Atlantic occurs in the surface layers. It is my understanding that in the Atlantic, the Subtropical Cells are integral to the overall MOC.

R2.22: We agree. Nevertheless, traditionally, most studies on AMOC variability using the maximum overturning transport below 300-m as the index to avoid the interference of the wind-driven subtropical cell. This is more proper for the purpose here, we think.

Reviewer #3 (Remarks to the Author):

This is my review of the paper entitled "Accelerated Slowdown of Atlantic Circulation Emerged in Optimal Fingerprint", submitted to

The paper relies on the assumption that the surface salinity in the South Atlantic can serve as a proxy for the AMOC changes, in that a decrease in the AMOC causes by anomalies in the subpolar North Atlantic would reflect in salinification of the South Atlantic. An index is then created, and compared against CMIP5/6 class models under different forcing scenarios. The paper states that this fingerprint has a better representation of the global warming signal than the previously used subpolar index.

The paper is well written, and makes an interesting/valid point. Salinity in the South Atlantic, for being an integrated variable, and less affected by surface fluxes and direct aerosol forcing, is a potential good fingerprint for ocean circulation. Salinity unfortunately suffers from lack of early data, and model biases. In addition, the South Atlantic does not only respond to the North Atlantic, but also has its own variability, since it is the link of three ocean basins. My comments go in this direction, and additional analysis/discussion is needed to guarantee the feasibility of this index under uncertainties and regional forcings.

Response to the overall comment: We appreciate the reviewer's insightful comments. Below are our detailed responses to the reviewer's questions. We copied the original review in blue and write our reply in black. Changes in the manuscript text are shown with red color highlighting.

Main comments:

1) It would be more informative to define an equation ($amoc \sim a*sal+B$, for example) for this relationship. Fig 1 was too qualitative, and we cannot really see this relationship there. This has been defined for the TNA index (Caesar et al., 2018). Correlations can be OK, but they do not show a clear view of the relationship of the magnitude of changes. I wonder if we can draw a unique relationship between the two indices.

R3.1: Thanks for the suggestion. The relationship between SSS index and AMOC is hard to

quantified during the historical period in models. First, there is a time lag (about one to two decades) between the southern AMOC and northern AMOC. Second, the time taken for southern AMOC change to produce a basin-mean change of salinity can be highly uncertain given the different ocean velocity in different models. These make modeled SSS change in response to AMOC change has a large spread in both time scale and magnitude. Only under strong warming climate after a long time period when the lag between the SSS index and AMOC becomes much smaller (Fig.3b), can the relationship be derived between the two with reasonable confidence. Such results have been shown in our previous paper (ref.35: Zhu and Liu 2020) which focused mainly on the warming scenario. Based on the trend correlations across warming scenarios ($R = 0.76$), we drew a quantitative relationship between the northern AMOC change and our SSS index change as $2.3 \pm 0.34 \text{ Sv}/0.1 \text{ psu}$ (Fig.3 in ref.35).

References:

Zhu, C., Liu, Z. Weakening Atlantic overturning circulation causes South Atlantic salinity pile-up. *Nature Climate Change* **10**: 998-1003(2020).

2) Trends in reanalyses and AMOC reconstructions:

I am a bit concerned about the discussion in lines 96-99. The authors claim that "this post-1980s AMOC weakening trend appears consistent with the AMOC reconstructions based on instrument measurements after ~1990s". First, a very small number of salinity observations were available in the South Atlantic until 2007, so most reanalysis suffer from the same bias from the Argo spinup since 2005. In addition, ref 33 state that: 1) the regions between 20 and 25S is dominated by the subtropical gyre, so the changes observed in that region may be just an expression of the wind stress curl changes in the region; 2) There is actually an increase trend of AMOC/MHT in 35S (Figure 14 in Ref 33), so I wonder why the present paper does not include 35S in the average estimate.

R3.2: Here “AMOC reconstructions” actually refer to the AMOC observations and reanalyses (the term “reconstruction” has also been used in the review paper of Jackson et al. 2022), not directly raw salinity measurement based reconstructions. To avoid the confusion, we have changed the word “reconstructions” to “*observations and reanalyses*” (Line 97). For data uncertainty, we admit that the salinity data in the South Atlantic is sparse before the Argo era and

we weakened our statement of long-term trend in the revised version by adding “likely”. Please see R3.11 and our reply to Reviewer 2 (R2.1) for more information. To address the meridional incoherency in the southern AMOC (SAMOC), we plotted the annual mean time series at 20°S, 25°S, 30°S and 35°S below. In our original manuscript, the time period for SAMOC is from 1993 to 2015 (2016 run) and now is extended to 2021 (2019 run). We can see that the SAMOCs at different latitudes consistently show a moderate weakening. Personal communications with Claudia Schmid from NOAA/AOSL (responsible for NOAA SAMOC) helped us learn more details about the SAMOC reconstructions. Compared with 2016 run, the 2019 run uses new reference velocity used to adjust the geostrophic velocity and the relationship between dynamic height and SSH is also recomputed. The results for 35°S and 25°S are consistent between the 2016 and 2019 run while results for 20°S and 30°S show more differences, indicating SAMOC reconstructions from Argo and Altimetry at those latitudes are more challenging. Dr. Schmid also suggested that at 20°S and 30°S the 2019 run is more robust than the old run. In the revised manuscript, the average estimate includes reconstructions from all the four latitudes from 1993 to 2021 using the 2019 run (Please see the new Fig.S2).

Fig.R3.1 NOAA SAMOC reconstructions at different latitudes.

3) Teleconnections to SA:

The South Atlantic is a place under influences of atmospheric teleconnections from the Pacific Ocean. Of particular importance here is the PDO, which has influences on long timescales in the South Atlantic (Lopez et al., 2016; Majumder et al., 2019, among others). So I wonder if part of the signal could be driven by teleconnections from the Pacific.

R3.3: Thanks for the insightful comment. It's true that the atmospheric teleconnections associated with internal variability, for example the PDO, can be a possible driver for SA change on interdecadal timescale. Our focus here, however, is actually the long-term (multidecadal to centennial) trend under anthropogenic forcing. These long term trend/variability are forced by external forcing, as shown in the MMEM in CMIP5 and CMIP6 experiments. (The MMEM filters out the internal variability, too). The lead-lag correlation between the northern AMOC and southern AMOC also supports a northern control. We have added some discussions in Line 255-261 to address this issue.

4) The Southern Annular Mode (SAM) effect on the Agulhas leakage and SA salinity has been explored in many papers (Marini et al., 2011, Goes et al., 2014, Durgadoo et al., 2013, Loveday et al., 2015, etc.), in which the southern hemisphere westerlies would force the changes in salinity, and later on potentially influencing the AMOC. SAM has suffered strong trends over the historical period. I wonder if this is the signal driving the changes in the SA. Indeed, these changes would bring SS anomalies in the same signal as the anomalies described in this paper, but via Agulhas Leakage high salinity input from the Indian Ocean.

R3.4: This is a very good question. Previous studies including those the reviewer mentioned show that the SH westerly has been strengthening and shifting southward (positive trend in SAM) over the historical period under global warming, which will input more saline water from the Indian Ocean. Therefore, the salinity pile-up in the South Atlantic can be caused by inter-basin exchange between the South Atlantic and Indian Ocean. We admit that in observations, this mechanism could be a candidate for salinity pile-up. From the model perspective, however, we think buoyancy-forced northern AMOC change is the major controlling factor for the salinity pile-up. There are several reasons. First, most coupled PI control simulations (including IPSL-CM6) show that northern AMOC weakening leads South Atlantic salinity pile-up by several

decades (Fig.S3), in contrast to the single model result (IPSL-CM4) reported in Marini et al. (2011). Second, in our ocean-alone sensitivity experiments described in Zhu and Liu (2020) and in R1.20, when surface forcing such as wind stress and E-P is fixed (and thus there is no SAM related changes), the salinity pile-up in the South Atlantic (relative to the Indo-Pacific) can be reproduced by a weakened AMOC alone. Third, we note that the roles of strengthening and southward shifting of SH westerlies would compete with each other, leaving Agulhas leakage not change too much (Durgadoo et al., 2013). We have added some discussions **around Line 255-261** to address this issue.

5) This is just a curiosity, since the authors have experience in this subject. According with the salt advection theory, does the model AMOC stability state influences if the South Atlantic gets saltier or fresher with the AMOC reduction? I asked this question following this statement in L.252 "Our study, combined with recent analysis that AMOC may have evolved from relatively stable period to a point close to critical transition"

R3.5: Good question. We think AMOC stability itself does not affect the surface salinity response in the South Atlantic when the AMOC is reduced. The key factor behind the salinity pile-up is the mean SSS gradient in the South Atlantic (the climatological salinity increases from the subpolar to subtropics). However, we do caution that further AMOC weakening may bring AMOC closer to a bistable state if we evaluate the stability of AMOC from a diagnostic indicator that is calculated as the overturning freshwater transport along 34°S in the Atlantic (see Dijkstra et al. 2011 for more details):

$$M_{\text{ovs}} = -\frac{1}{S_0} \int_{34S}^z \bar{v}' (\langle S \rangle - S_0) dz$$

Most climate models show a stable AMOC with positive M_{ovs} (Table R3.1; Zhu et al., 2018). As shown in our previous paper (Zhu and Liu, 2020), the salinity pile-up in the South Atlantic is projected to intensify with global warming and further AMOC weakening (for example, under RCP4.5 scenario). The salinification in the South Atlantic would reduce M_{ovs} or even reverse it (Table R3.1), favoring a less stable AMOC.

Tabel R3.1 Mobs in historical period and RCP4.5 scenario in 8 CMIP5 models.

Climate model	Mobs_Hist	Mobs_RCP4.5
BCC-CSM1.1	0.10	0.01
CanESM2	0.16	0.08
CESM1-CAM5	0.00	0.00
CNRM-CM5	0.04	0.01
GFDL-ESM2G	0.15	-0.02
HadGEM2-ES	0.16	0.11
IPSL-CM5A-LR	-0.14	-0.19
MIROC-ESM	-0.01	-0.03

References:

Drijfhout, S. S., Weber, S., van der Swaluw, E. 2011: The stability of the MOC as diagnosed from model projections from pre-industrial, present and future climates. *Clim. Dyn.* **37**(7–8):1575–1586.

Zhu, C., Liu, Z. & Gu, S. 2018: Model bias for South Atlantic Antarctic intermediate water in CMIP5. *Clim. Dyn.* **50**: 3613-3624.

Other comments:

Fig 1a- why are the timeseries stopping in 2015? Most of these indices could be extended further on to the present.

R3.6: We have extended the observational timeseries in the revised Fig.1 and Fig.S2. For models, the end years of historical simulations are 2005 and 2014 for CMIP5 and CMIP6, respectively.

Fig 3 - It is really hard to see the difference between red and orange dots.

Green and black correlations (TNA and SAMOC respectively) are much more stable than red and orange.

R3.7: As we mentioned in the text, the two SSS indices (S_SA in orange and S_S in red) behave similarly. Our SSS indices have a better performance under external forcing, such as

anthropogenic GHGs and aerosol forcing, compared with those under internal variability.

L.89 -Typo: Highly "uncertain"

R3.8: Corrected.

L. 115-117 The two indices should indeed behave similarly because salinity is more or less conserved globally, different than temperature.

R3.9: We agree.

L.120 "In contrast to all previous AMOC fingerprints, our SSS index is unique in that it represents the remote AMOC response to the buoyancy flux in the SPNA deep-convection region, which is known to force AMOC signal propagating southward coherently (35-37)" I do not know how the authors arrived in this conclusion. It is not clear to me what forces what. Is SSS responding to the AMOC weakening or forcing it? It would be good to show some meridional hovmollers of salinity to define this propagation.

R3.10: Here "which" refers to "the buoyancy flux in the SPNA deep-convection region". We have rephrased the sentence, as "*In contrast to all previous AMOC fingerprints that are located locally in the North Atlantic, our SSS index is unique in that it represents a remote AMOC response to the buoyancy flux forcing in the SPNA deep-convection region, which is known to force AMOC signal propagating southward coherently*" (Line 116-119).

L.112-125 "For the real world, the observed S_s or S_{SA} shows a slow decreasing trend in EN4 data prior to 1990s (albeit uncertain in the Ishii data, which is too short), but a clearly accelerated weakening after 1990s in both EN4 and Ishii data sets (Fig.1b,c, Fig.S2a)." How significant is this? There is very little salinity data in the South Atlantic previous to 2005 (See main comment 2).I would suggest the authors to look at the "coverage weights", which is provided by EN4 dataset to infer some significance of the data trends, particularly prior to 1990.

R3.11: Thanks again for pointing the data issue. We have checked the profile numbers and coverage weights (see our reply to Reviewer 2 R2.1 and Fig.R2.1). As we replied in R2.1, there are rare salinity observations before 1950s and the data becomes plentiful only after 2000s (Argo era). In the revised manuscript, we have weakened the statement of salinity trend on longer time scale in title and many places in the text by adding "likely". Also, we have checked another

salinity data from Institute of Atmospheric Physics (IAP), which is shown to be “more reliable for examining long-term salinity changes” (Cheng et al., 2020). The SSS index using IAP salinity data is shown in Fig.R2.2 along with 90% CI (sampling error). The decreasing trend after the 1970s in IAP SSS index is shown to be significant, with change after the 1990s consistent with EN4 and Ishill dataset. Please see R2.1 for more information. We also note that, in the early period, our salinity index is similar to the T_NA index (which indeed has more data available), both showing little variability. Therefore, the lack of AMOC variability in the earlier period seems to be consistent in these two independent indices. We have added some discussion on the data issue in Line 129-132.

References:

Cheng, L., et al.2020: Improved Estimates of Changes in Upper Ocean Salinity and the Hydrological Cycle. *J.Clim.*, **33**:10357-10381.

L. 174-176 The statement that "S_s is forced completely by the AMOC change, instead of surface E-P forcing associated to the hydrology response" is not shown correctly in Figures S4 and S5. E-P should be associated to salinity changes (dS/dt) not a direct relationship as shown in Figure S4. To analyze it correctly, S_s should be compared to the time integral of E-P (dS~(E-P)*dt).

R3.12: We agree and thanks for pointing that out. In the revision (Fig.S4) and here the Fig.R3.2, S_s is compared to the time integral of (E-P). It is found that (E-P) change dominates local STSA SSS (S_{SA}) change at a rate of 2m/0.1psu (Fig.R3.2c,d). This scaling relationship can be also seen in Hist-GHG and Hist-aers. For S_s however, although historical changes in S_s is comparable to change in S_{SA}, the scaling relationship reduces to 1m/0.1psu, indicating that (E-P)_s change only accounts for about half of the S_s change (Fig.R3.2a,b). The remaining 50% is likely attributed to circulation change (AMOC change). Moreover, the scaling relationship between accumulated (E-P)_s and S_s in Hist-GHG and Hist-aer is reversed to -0.2m/0.1psu, indicative of a small negative effect of (E-P)_s on S_s. Therefore, we argue that S_s in Hist-GHG and Hist-aer “is forced predominantly by the AMOC change, instead of surface E-P forcing associated to the hydrology response”. For Fig.S5, we mainly focus on the signs instead of the exact values. Changes in the time integral of (E-P) generally have the same sign as the linear trend of (E-P). The dominant

role of AMOC change on S_s over (E-P) under global warming has also been confirmed in model experiments using salinity budget analysis in Zhu and Liu (2020).

Fig.R3.2 Response of S_{SA}/S_s and accumulated $(E-P)_{SA}/(E-P)_s$ in historical, Hist-GHG and Hist-aer simulations. Anomalies are relative to the means of 1850-1900. Note the neglectable role of $(E-P)_s$ on S_s especially under GHGs and aerosol forcing. Here $(E-P)_s$ is the (E-P) change of STSA relative to STSIP, defined similar to S_s .

References:

Zhu, C., Liu, Z. Weakening Atlantic overturning circulation causes South Atlantic salinity pile-up. *Nature Climate Change* **10**: 998-1003(2020).

Figure S3: As far as these simulations go, the NAMOC precedes the SAMOC in most simulations. The correlations with S_{SA} are in general small, although according to the authors they are significant. It is really strange that correlations of ~ 0.05 are statistically significant. Am I missing something? It deserve explanation.

R3.13: In FigureS3, the significance level is determined by Monte Carlo method (sampling

10000 times). Time series with longer length, such as those for CanESM5(1000yr) and IPSL-CM6A-LR(1200 year), generally have smaller correlation values at a given significance level.

References:

Marini, C., Frankignoul, C., & Mignot, J. (2011). Links between the Southern Annular Mode and the Atlantic Meridional Overturning Circulation in a Climate Model, *Journal of Climate*, 24(3), 624-640.

Goes, M., Wainer, I., and Signorelli, N. (2014), Investigation of the causes of historical changes in the subsurface salinity minimum of the South Atlantic, *J. Geophys. Res. Oceans*, 119, 5654–5675, doi:10.1002/2014JC009812.

Durgadoo, J. V., B. R. Loveday, C. J. C. Reason, P. Penven, and A. Biastoch (2013), Agulhas leakage predominantly responds to the Southern Hemisphere westerlies, *J. Phys. Oceanogr.*, 43, 2113– 2131.

Loveday, B. R., Penven, P., and Reason, C. J. C. (2015), Southern Annular Mode and westerly-wind-driven changes in Indian-Atlantic exchange mechanisms. *Geophys. Res. Lett.*, 42, 4912–4921. doi: 10.1002/2015GL064256.

Lopez, H., Dong, S., Lee, S.-K., & Campos, E. (2016). Remote influence of interdecadal Pacific oscillation on the South Atlantic meridional overturning circulation variability. *Geophysical Research Letters*, 43, 8250– 8258. <https://doi.org/10.1002/2016GL069067>

Majumder, S., Goes, M., Polito, P. S., Lumpkin, R., Schmid, C., & Lopez, H. (2019). Propagating modes of variability and their impact on the western boundary current in the South Atlantic. *Journal of Geophysical Research: Oceans*, 124, 3168–3185. <https://doi.org/10.1029/2018JC014812>

REVIEWER COMMENTS

Reviewer #1 (Remarks to the Author):

Review of Likely Accelerated Weakening of Atlantic Overturning Circulation Emerged in Optimal Fingerprint Past and future response to the North Atlantic warming hole to anthropogenic forcing by Zhu et al.

I appreciate the work the authors have done to accommodate the reviewers' comments, and I think that these changes have improved the manuscript. I believe the manuscript is close to being acceptable, but I would like for the following three points to be addressed/mentioned in the paper.

1) As it only detects its long-term trend, the salinity index cannot be used as an indicator for annual or even interannual AMOC variability. It also can not give an estimate of the actual slowdown in the volume transport.

2) The SS index is not consistent with the AMOC strengthening observed in the reconstructions of the subtropical AMOC in the North Atlantic in the 80s (the authors say this is because this strengthening is due to internal variability, which might be, it just should be mentioned in the script).

3) Due to sparse data availability the SS index before the 1960s is probably not that meaningful. I think the plot containing the observation weights has to be included in the final paper (at least as a subplot).

Reviewer #2 (Remarks to the Author):

This is a re-review of the manuscript. I thought the authors did a good job of addressing my prior concerns. The manuscript presents an interesting alternative perspective on Atlantic Meridional Overturning Circulation trends. The work is timely, on a controversial topic, and substantially contributes to bettering our understanding of AMOC trends. I think the methods are sound. The revised manuscript has appropriate caveats and better-explained reasoning.

Reviewer #3 (Remarks to the Author):

This is my review of the revised manuscript "Accelerated Slowdown of Atlantic Circulation Emerged in Optimal Fingerprint".

The paper describes an optimal fingerprint based on salinity in the South Atlantic, which is less susceptible by decadal noise than the ones in the North for being filtered by the equatorial "buffer" (as in Johnson and Marshall, 2004). The text improved considerably from the previous version, and the authors responded to all my previous concerns. So in my opinion, the paper is potentially publishable after some additional clarifications.

I still have a couple questions related to the significance of some correlations and the impact of the atmospheric forcing. In addition, the paper should highlight more clearly in the introduction that some of the questions addressed are related to their previous paper (Zhu et al., 2020).

Main comments:

Introduction:

1) It should be made clear that his paper is a "continuation" of Zhu et al., (2020) with deeper analysis containing single forcing models and observations. This would influence, for example this sentence in the abstract: L.41 "Here, for the first time we present observational and modeling evidences for a likely accelerated weakening of AMOC since the 1980s under the combined forcing of anthropogenic greenhouse gases and aerosols", which has been modified from the previous paper: "Here we show observational and modelling evidence of a remote indicator of AMOC slowdown outside the North Atlantic"

Significance of the correlations:

2) In Fig 2c, L. 109-110: The correlation of S_SA and AMOC in the CMIP6 models (0.58; dots in Fig

2c) seems to be driven solely by the high leverage of the GISS model. Thus, the sentence in L. 109-110 "Therefore, TNA cannot serve as a reliable fingerprint for long-term forced AMOC response.", which is based on this figure, has to be taken with a grain of salt.

This correlation is calculated only by 9 points, and all the other data points except the GISS model are very similar to Figure 2a, which shows a much smaller correlation of 0.22. Therefore, it should be tested if the GISS model is really driving the correlation. This test could be made by excluding that point in the correlation. If the correlation remains close to the initial one, it should be considered robust, otherwise it may draw a red flag. Actually, a better analysis should be comparing, for each model, the correlation of the ensemble means and the means of the correlations of the ensemble members, instead of the correlation of the ensemble means (Fig 2c) and each ensemble separately (Fig 2d).

Impact of the surface (E-P) fluxes:

3) Fig. S4, S5 and L.184-185. Fig S4 now presents integrated fluxes over time, which is more consistent with the salinity budget (ΔS and $(E-P) \cdot dt$). However, because they are in different units (PSU against Meter), it is still hard to understand what the main driver of the salinity anomalies is. If presented in the same units, it would be useful to understand the amount of forcing from the atmosphere.

In addition, from Figure S5, the relationship in the South Atlantic is direct (salinification/increased evaporation and freshening/increase precipitation). Consequently, in Fig S4a,b, the lack of relationship between S_s and $(E-P)_s$ seems to arise almost solely from the Indian Ocean, not from the Atlantic. This relationship with the surface fluxes should be better presented and explained.

Minor Comments:

Fig S1 The CDF the panel (b) is not described in the caption. Also use in "the" combined

Fig. 5 Should the unit in Panel be W/m^2 ?

L.185 "hydrological" response

Response to Reviews

Reviewer #1 (Remarks to the Author):

Review of Likely Accelerated Weakening of Atlantic Overturning Circulation Emerged in Optimal Fingerprint by Zhu et al.

I appreciate the work the authors have done to accommodate the reviewers' comments, and I think that these changes have improved the manuscript. I believe the manuscript is close to being acceptable, but I would like for the following three points to be addressed/mentioned in the paper.

We appreciate the reviewer's positive feedback and further valuable comments. Below are our point-by-point responses.

1) As it only detects its long-term trend, the salinity index cannot be used as an indicator for annual or even interannual AMOC variability. It also can not give an estimate of the actual slowdown in the volume transport.

R1.1 It's true that our salinity index is more preferable for detecting long-term trend, which is more related to anthropogenic forcing. This time scale issue has been mentioned in many places of the text. As we replied in R3.1 in the 1st *Response to Reviews*, the relationship between SSS index and AMOC is hard to quantified during the historical period in models. Only under strong warming climate after a long time period, can the relationship be derived between the two with reasonable confidence. Such results have been shown in our previous paper (ref.35: Zhu and Liu 2020) which focused mainly on the warming scenario. Based on the trend correlations across warming scenarios ($R = 0.76$), we drew a quantitative relationship between the northern AMOC change and our SSS index change as $2.3 \pm 0.34 \text{ Sv}/0.1 \text{ psu}$ (Fig.3 in ref.35).

References:

Zhu, C., Liu, Z. Weakening Atlantic overturning circulation causes South Atlantic salinity pile-up. *Nature Climate Change* **10**: 998-1003(2020).

2) The SS index is not consistent with the AMOC strengthening observed in the reconstructions of the subtropical AMOC in the North Atlantic in the 80s (the authors say this is because this strengthening is due to internal variability, which might be, it just should be mentioned in the script).

R1.2 For reconstructions of the *subtropical* AMOC as investigated in Jackson et al. (2022), “*there is some evidence of the AMOC strengthening from 2001 to 2005 and strong evidence of a weakening from 2005 to 2014. Such large interannual and decadal variability complicates the detection of ongoing long- term trends, but does not preclude a weakening associated with anthropogenic warming*”. It seems that only subpolar T_NA fingerprint shows strong strengthening since 1990s, which is likely caused by interdecadal variability, notably the AMO. This strengthening has been fully discussed in our paper (Line 68-72). Please also see R1.13 in the 1st *Response to Reviews* for more related information.

References:

Jackson, L. C., et al. The evolution of the North Atlantic Meridional Overturning Circulation since 1980. *Nat. Rev. Earth. Environ.* (2022). <https://doi.org/10.1038/s43017-022-00263-2>.

3) Due to sparse data availability the SS index before the 1960s is probably not that meaningful. I think the plot containing the observation weights has to be included in the final paper (at least as a subplot).

R1.3 Thanks for the suggestion. The observation weight has been added as a subplot in the new Fig.1 (also can be seen below).

Reviewer #2 (Remarks to the Author):

This is a re-review of the manuscript. I thought the authors did a good job of addressing my prior concerns. The manuscript presents an interesting alternative perspective on Atlantic Meridional Overturning Circulation trends. The work is timely, on a controversial topic, and substantially contributes to bettering our understanding of AMOC trends. I think the methods are sound. The revised manuscript has appropriate caveats and better-explained reasoning.

We appreciate the reviewer's positive feedback. The reviewer's constructive comments and suggestions in the first-round review have significantly improved the presentation of our manuscript.

Reviewer #3 (Remarks to the Author):

This is my review of the revised manuscript "Accelerated Slowdown of Atlantic Circulation Emerged in Optimal Fingerprint".

The paper describes an optimal fingerprint based on salinity in the South Atlantic, which is less susceptible by decadal noise than the ones in the North for being filtered by the equatorial "buffer" (as in Johnson and Marshall, 2004). The text improved considerably from the previous version, and the authors responded to all my previous concerns. So in my opinion, the paper is potentially publishable after some additional clarifications.

I still have a couple questions related to the significance of some correlations and the impact of the atmospheric forcing. In addition, the paper should highlight more clearly in the introduction that some of the questions addressed are related to their previous paper (Zhu et al., 2020).

We appreciate the reviewer's positive feedback and further constructive comments. Below are our point-by-point responses to the additional concerns.

Main comments:

Introduction:

1) It should be made clear that his paper is a "continuation" of Zhu et al., (2020) with deeper analysis containing single forcing models and observations. This would influence, for example this sentence in the abstract: L.41 "Here, for the first time we present observational and modeling evidences for a likely accelerated weakening of AMOC since the 1980s under the combined forcing of anthropogenic greenhouse gases and aerosols", which has been modified from the previous paper: "Here we show observational and modelling evidence of a remote indicator of AMOC slowdown outside the North Atlantic"

Significance of the correlations:

R3.1 Thanks for the suggestion. In our previous paper (Zhu and Liu, 2020), we mainly focused on warming scenarios and showed that a weakening AMOC can lead to salinity pile-up in the South Atlantic (relative to the South Indo-Pacific). In that study, we did not make a judgement between our salinity pile-up index and the classic warming hole index. In the present study, we show that the salinity index can serve as an *optimal* fingerprint for detecting long-term forced AMOC change in comparison with the warming hole index. Based on this optimal salinity index, we further suggest a likely accelerated weakening of AMOC since the 1980s under the combined anthropogenic forcing. Nevertheless, the point of this review is considerably taken. In abstract, we have deleted “for the first time” in L.41. Also, in the introduction of this salinity fingerprint, we have modified the L.111-113 to “Alternatively, we will show below that a salinity-based fingerprint firstly proposed in our previous study³⁵ can be a better choice for detecting this long-term forced AMOC change...”

2) In Fig 2c, L. 109-110: The correlation of S_SA and AMOC in the CMIP6 models (0.58; dots in Fig 2c) seems to be driven solely by the high leverage of the GISS model. Thus, the sentence in L. 109-110 "Therefore, TNA cannot serve as a reliable fingerprint for long-term forced AMOC response.", which is based on this figure, has to be taken with a grain of salt.

This correlation is calculated only by 9 points, and all the other data points except the GISS model are very similar to Figure 2a, which shows a much smaller correlation of 0.22. Therefore, it should be tested if the GISS model is really driving the correlation. This test could be made by excluding that point in the correlation. If the correlation remains close to the initial one, it should be considered robust, otherwise it may draw a red flag. Actually, a better analysis should be comparing, for each model, the correlation of the ensemble means and the means of the correlations of the ensemble members, instead of the correlation of the ensemble means (Fig 2c) and each ensemble separately (Fig 2d).

R3.2 Our argument of “TNA cannot serve as a reliable fingerprint for long-term forced AMOC response” is actually based on the comparison between Fig.2e and Fig.2f. Therefore, the outlier model of HadGEM3 (not GISS model) in Fig.2c does not affect the argument. We showed that the high cross-member (with strong internal variability) trend correlation between TNA and

AMOC (~0.6) is largely reduced (even being nearly 0 for CMIP5 models) in the cross-model (more forced signal) trend correlation, indicating that T_{NA} fingerprint is less reliable for representing the forced AMOC change. In contrast, for both salinity indices, the cross-model trend correlations are enhanced compared with their corresponding cross-member trend correlations (Fig.2a vs. Fig.2b and Fig.2c vs. Fig.2d) and are also larger than those of the T_{NA} (in Fig.2c, the correlations for CMIP6 with and without HadGEM3 model are 0.58 and 0.55, respectively; in Fig.2e, the correlations for CMIP6 with and without HadGEM3 model are 0.34 and 0.17, respectively).

Impact of the surface (E-P) fluxes:

3) Fig. S4, S5 and L.184-185. Fig S4 now presents integrated fluxes over time, which is more consistent with the salinity budget (ΔS and $(E-P) \cdot dt$). However, because they are in different units (PSU against Meter), it is still hard to understand what the main driver of the salinity anomalies is. If presented in the same units, it would be useful to understand the amount of forcing from the atmosphere.

In addition, from Figure S5, the relationship in the South Atlantic is direct (salinification/increased evaporation and freshening/increase precipitation). Consequently, in Fig S4a,b, the lack of relationship between S_s and (E-P)_s seems to arise almost solely from the Indian Ocean, not from the Atlantic. This relationship with the surface fluxes should be better presented and explained.

R3.3 We thank the reviewer for these comments. It is clear that the contribution of (E-P) is opposite to the S_s change under anthropogenic aerosol/greenhouse gas forcing (Fig.S4), indicating that S_s change is more driven by oceanic circulation change. In the future with combined and strengthening anthropogenic forcing, the role of circulation change can be more dominant. These are our main points and can be qualitatively seen from Fig.S4.

Quantitatively, the contribution of (E-P) to salinity change can be evaluated as $\int_0^t \frac{(E-P) \times S_0}{H_0} dt$,

where H₀ is the layer thickness and S₀ is the reference salinity (often taken as 34.7 psu). To better illustrate the individual roles of surface (E-P) flux and ocean circulation, in our previous paper (Zhu and Liu, 2020), we calculate the thermocline (upper 200~300m) salinity budget for both coupled simulation and ocean alone simulation under warming scenarios (Extended Data

Fig.7,8&9). Our sensitivity experiments clearly show that pure E-P forcing (derived from RCP45 anomaly) forces nearly equal salinification between the South Indo-Pacific and the South Atlantic; only with a weakening AMOC, can the thermocline salinity pile-up occur in the South Atlantic relative to the Indo-Pacific (Extended Data Fig.8&9 in Zhu and Liu 2020). This relationship is further confirmed in a coupled CESM simulation, in which although the (E-P) forcing is stronger in the South Indo-Pacific Ocean, the salinity increase is more profound in the South Atlantic due most likely to the AMOC change (Extended Data Fig.7 in Zhu and Liu 2020). The decoupling between (E-P)_S and S_S under strong anthropogenic forcing (GHGs and aerosols) has been discussed in Supplementary Text and readers are referred to our previous paper for detailed salinity budget analysis. Here for multi-model ensemble mean (MMEM) salinity change, we did not carry out such salinity budget analysis since the signals of (E-P)_S and S_S can be from different models especially for historical run. Since the surface salinity signal is consistent with the subsurface signal in our study region (please see our previous paper for zonal-vertical distribution of salinity change) and the surface salinity observation is more reliable, we have mostly focused on the surface change in the present study.

We note that both (E-P) and AMOC change can influence salinity in the South Atlantic. That is, global warming causes AMOC weakening and subtropical salinification (as part of hydrological response), both leading to an increase in S_{SA}. In contrast, by subtracting S_{STSIP} from S_{SA}, the S_S index is less affected by the quasi-zonally uniform (E-P) influence (such influence can be seen by comparing Fig.S4a and S4c). Therefore, the seemingly good relationship between (E-P)_{SA} and S_{SA} cannot preclude the effects of AMOC change while the lack of relationship between (E-P)_S and S_S indeed highlights the role of the AMOC.

References:

Zhu, C., Liu, Z. Weakening Atlantic overturning circulation causes South Atlantic salinity pile-up. *Nature Climate Change* **10**: 998-1003(2020).

Minor Comments:

Fig S1 The CDF the panel (b) is not described in the caption. Also use in "the" combined

R3.4 Modified.

Fig. 5 Should the unit in Panel be W/m^2 ?

R3.5 Modified. It should be W/m^2 .

L.185 "hydrological" response

R3.6 Modified.